# Causal role of the frontal eye field in attention-induced ocular dominance plasticity

**Fangxing Song**[1,2], **Xue Dong**[1,2]*, **Jiaxu Zhao**[1,2], **Jue Wang**[1,2], **Xiaohui Sang**[1,2], **Xin He**[1,2], **Min Bao**[1,2]*

[1]CAS Key Laboratory of Behavioral Science, Institute of Psychology, Chinese Academy of Sciences, Beijing, China; [2]Department of Psychology, University of Chinese Academy of Sciences, Beijing, China

*For correspondence:
dongx@psych.ac.cn (XD);
baom@psych.ac.cn (MB)

**Competing interest:** The authors declare that no competing interests exist.

**Abstract** Previous research has found that prolonged eye-based attention can bias ocular dominance. If one eye long-termly views a regular movie meanwhile the opposite eye views a backward movie of the same episode, perceptual ocular dominance will shift towards the eye previously viewing the backward movie. Yet it remains unclear whether the role of eye-based attention in this phenomenon is causal or not. To address this issue, the present study relied on both the functional magnetic resonance imaging (fMRI) and transcranial magnetic stimulation (TMS) techniques. We found robust activation of the frontal eye field (FEF) and intraparietal sulcus (IPS) when participants were watching the dichoptic movie while focusing their attention on the regular movie. Interestingly, we found a robust effect of attention-induced ocular dominance shift when the cortical function of vertex or IPS was transiently inhibited by continuous theta burst stimulation (cTBS), yet the effect was significantly attenuated to a negligible extent when cTBS was delivered to FEF. A control experiment verified that the attenuation of ocular dominance shift after inhibitory stimulation of FEF was not due to any impact of the cTBS on the binocular rivalry measurement of ocular dominance. These findings suggest that the fronto-parietal attentional network is involved in controlling eye-based attention in the 'dichoptic-backward-movie' adaptation paradigm, and in this network, FEF plays a crucial causal role in generating the attention-induced ocular dominance shift.

## eLife assessment

This **important** study combines psychophysics, fMRI, and TMS to reveal a causal role of FEF in generating an attention-induced ocular dominance shift, with potential relevance for clinical applications. The evidence supporting the claims of the authors is **convincing**. The work will be of broad interest to perceptual and cognitive neuroscience.

## Introduction

Selective attention allows for the selection of pertinent information from a vast array of input for further cognitive processing (*Treisman, 1969*; *Wolfe et al., 1989*). It has been well known that attentional selection can be location-specific (*Posner, 1980*), feature-specific (*Corbetta et al., 1990*), or object-specific (*Egly et al., 1994*). Notably, selective attention can also be based on the eye of origin for visual input, named eye-based attention (*Neisser and Becklen, 1975*). For instance, presenting top-down attentional cues to one eye can intensify the competition strength of input signals in the

**Figure 1.** Schematic diagram of the (**A**) binocular rivalry task, (**B**) 'dichoptic-backward-movie' paradigm, and (**C**) blob target stimulus (see the gray region around the mouth in this example). This figure presents identifiable images of human faces solely for the purpose of demonstration, which were captured from the authors (F. Song, J. Wang, and J. Zhao) of this article. The movie images used in the experiment are not displayed in this figure due to potential copyright issues.

attended eye during binocular rivalry (**Choe and Kim, 2022**; **Zhang et al., 2012**) and shift the eye balance towards the attended eye (**Wong et al., 2021**).

It is worth noting that eye-based attention not only exerts real-time effects on information processing (**Choe and Kim, 2022**; **Wong et al., 2021**; **Zhang et al., 2012**), but also exhibits a counterintuitive perceptual aftereffect that prolonged attention to a monocular pathway (or eye-based attention) can result in a shift of ocular dominance towards the unattended eye (**Song et al., 2023**; **Wang et al., 2021**). In **Song et al., 2023**'s 'dichoptic-backward-movie' adaptation paradigm (see **Figure 1B**), participants were presented with regular movie images in one eye (i.e. attended eye) while the other eye (i.e. unattended eye) received the backward movie images of the same episode. They were also instructed to try their best to follow the logic of the regular movie and ignore the superimposed backward movie. Therefore, the goal-directed eye-based attention was predominantly focused on the attended eye. **Song et al., 2023** found that the predominance of the unattended eye in binocular rivalry increased after 1 hr of adaptation to the 'dichoptic-backward-movie,' indicating a shift of perceptual ocular dominance towards the unattended eye. Since the overall energy of visual input from the two eyes was balanced throughout the adaptation period, the change of ocular dominance after adaptation is thought to result from unbalanced eye-based attention rather than unbalanced input energy as in typical short-term monocular deprivation (**Bai et al., 2017**; **Lunghi et al., 2011**; **Zhou et al., 2014**). In short-term monocular deprivation, an input signal from one eye is blocked. Accordingly, attention is biased towards the non-deprived eye. However, it is difficult to tease apart the potential contribution of unbalanced eye-based attention from the consequence of the unbalanced input energy, as the deprived eye is also the unattended eye. Therefore, the advantage of the 'dichoptic-backward-movie' adaptation paradigm is to balance the input energy across the eyes but leave attention unbalanced across the eyes. Furthermore, the steady-state visually evoked potentials (SSVEPs) elicited by the adapting stimuli in each eye were observed in both the occipital and frontal sites (**Song et al., 2023**), with a significant positive correlation between the frontal neural-activity index (a normalized SSVEP response related to selective attention) and the shift of perceptual ocular dominance. Their findings imply the contribution of frontal attentional system in producing such an adaptation effect.

However, this account is based on the correlation result rather than causal evidence. Besides, the frontal activation observed by *Song et al., 2023* is primarily located in the frontal pole, which is not considered a key area directly responsible for attentional control (*Iidaka, 2017*; *Jääskeläinen et al., 2016*; *Leminen et al., 2020*; *Xie et al., 2018*). As SSVEP is limited to detecting brain activities that oscillate at the flickering frequency (and its harmonics) of the visual stimulus, brain activities that do not precisely align with the flickering frequency may be missed (*Norcia et al., 2015*). Consequently, the activities of typical brain regions associated with the attentional network, such as the parietal and dorsolateral prefrontal cortex (*Vossel et al., 2014*), may not be captured by the SSVEP measures.

Therefore, the present study combines the use of fMRI and TMS to further investigate the neural mechanisms underlying the perceptual aftereffects led by 'dichoptic-backward-movie' adaptation, seeking direct causal evidence for the role of eye-based attention in modulating ocular dominance. In Experiment 1, we used fMRI to identify any brain regions associated with eye-based attention when participants were watching the dichoptic movie. Subsequently, in Experiment 2, we examined whether transiently suppressing the cortical function of the identified brain regions using cTBS (*Huang et al., 2005*) would impair the formation of ocular dominance shift induced by 'dichoptic-backward-movie' adaptation. Participants watched the dichoptic movie for 50 min after the delivery of cTBS, with their ocular dominance measured before and after adaptation by a binocular rivalry task. The fundamental logic of Experiment 2 can be summarized as follows. First, the brain regions identified in Experiment 1 are related to eye-based attentional control, but do not necessarily contribute to the formation of ocular dominance shift. Second, if any of the identified brain regions play a causal role in the attention-induced ocular dominance plasticity, transiently suppressing its cortical function will sharply weaken the magnitude of ocular dominance shift as compared to delivering the cTBS over the vertex, a common control site in TMS studies. Thirdly, for a brain region with a causal role in generating the ocular dominance shift, whether suppression of its cortical function can completely eliminate the effect of ocular dominance shift or not may depend on the depth or efficiency of the suppression led by cTBS. In other words, even if the effect of ocular dominance shift does not vanish after the brain region is suppressed, as long as the effect becomes weakened, the brain region is still thought to play a causal role in the attention-induced ocular dominance plasticity. Experiments 3 and 4 further confirmed that the delivery of cTBS affected the formation of attention-induced ocular dominance plasticity during adaptation rather than either the performance of binocular rivalry task or the visibility of monocular stimuli.

## Results

### Experiment 1: The function role of fronto-parietal areas in controlling eye-based attention

We first tried to identify any brain regions associated with eye-based attention using fMRI in Experiment 1. During the experimental runs of fMRI scanning, participants viewed movie images either dichoptically (*Figure 2A*) or binocularly (*Figure 2B*) through the paper red-blue stereoscopic glasses. In the dichoptic condition, regular movie images were presented to one eye while the other eye received the backward movie images of the same episode. In the binocular condition, the same regular movie images were presented to both eyes. Participants were instructed to attentively watch the regular movie and comprehend the plot. Besides, we compared the response difference between dichoptic and binocular conditions in the experimental runs with that in the control runs, where participants performed a central rapid serial visual presentation (RSVP) task while watching the same movie stimuli as those of the experimental runs (*Figure 2C*), to eliminate potential interference arising from different visual inputs between the dichoptic and binocular conditions.

### Behavior result

After each experimental run, participants were required to answer a question about the movie plot. The results showed a high response accuracy to the questions ($M$=91.67%, SD = 14.91%), indicating that the participants had paid close attention to the regular movie and clearly understood the plot.

In the control runs, participants also exhibited a reasonable performance in the RSVP task ($d'$=3.33, SD=0.83). A 2 (experiment condition: dichoptic, binocular)×3 (run: run1, run2, run3) repeated measurements analysis of variance (ANOVA) was conducted on the d-prime score, which indicated

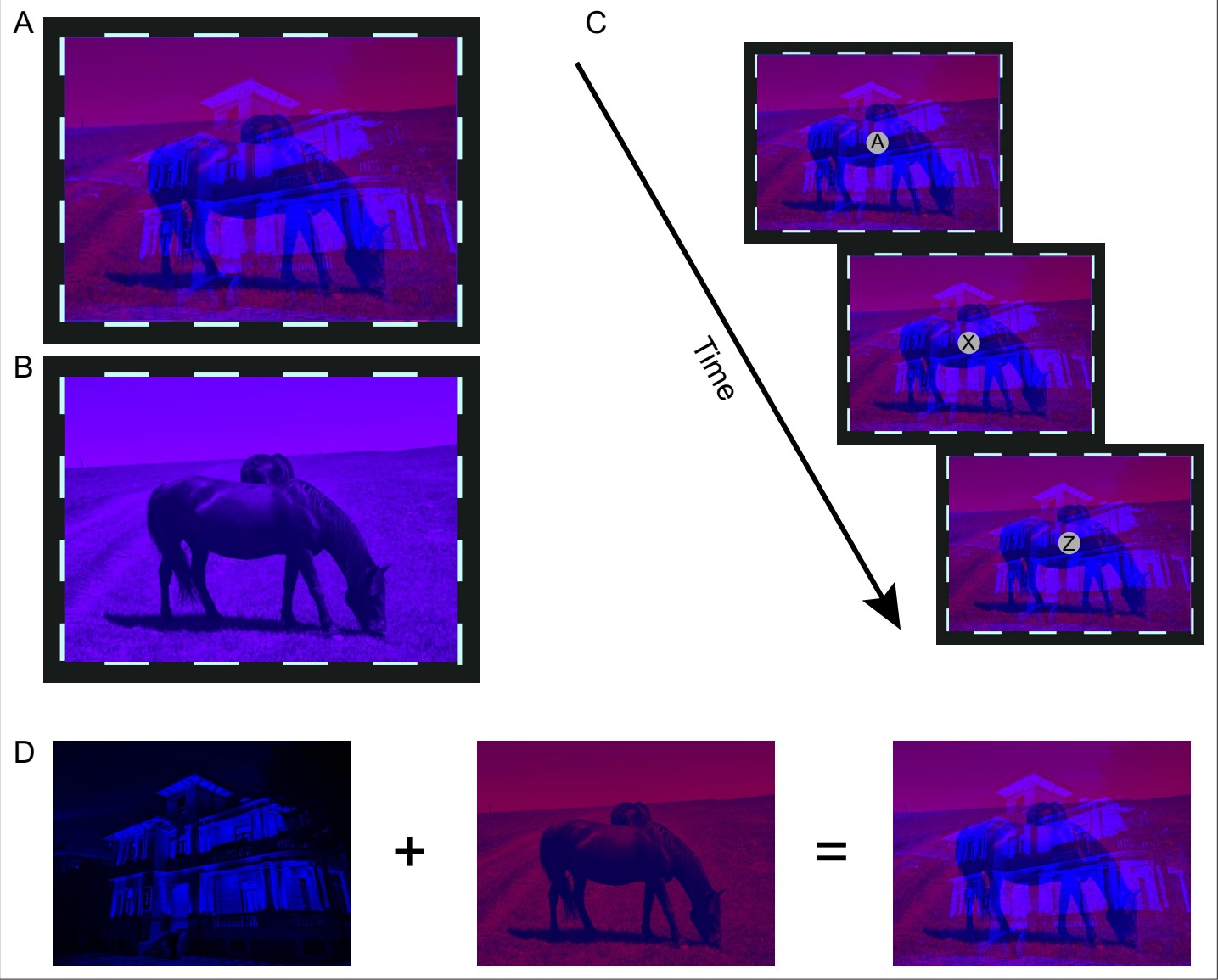

**Figure 2.** The schematic of stimuli in (**A**) dichoptic condition or (**B**) the binocular condition in the experimental runs. (**C**) The schematic of rapid serial visual presentation (RSVP) task in the control runs. For demonstration purposes, the letters and fixation points are enlarged. (**D**) Schematic illustration of the generation process for red-blue movie images employed in the dichoptic condition.

no significant main effect or interaction ($p$s >0.09). The generally favorable RSVP performance across both experimental conditions suggested that participants faithfully focused their attention on the central letter stream without being much affected by the movie stimuli in the control runs. Otherwise, the RSVP performance would be found better in the binocular condition than in the dichoptic condition, because a disobedient participant who liked to complete the RSVP task and watch a movie simultaneously had to pay more attention to the regular movie images (thus less attention to the letter stream) in the dichoptic condition than in the binocular condition given the annoying bistable nature of dichoptic stimulus presentation.

## fMRI result

It was only in the dichoptic condition of experimental runs that participants had to selectively pay more attention to one eye (i.e. eye-based attention). Therefore, we speculate that if certain brain regions exhibit greater activities in the dichoptic condition as compared to the binocular condition in the experimental runs but not in the control runs, the activation of these brain regions could be attributable

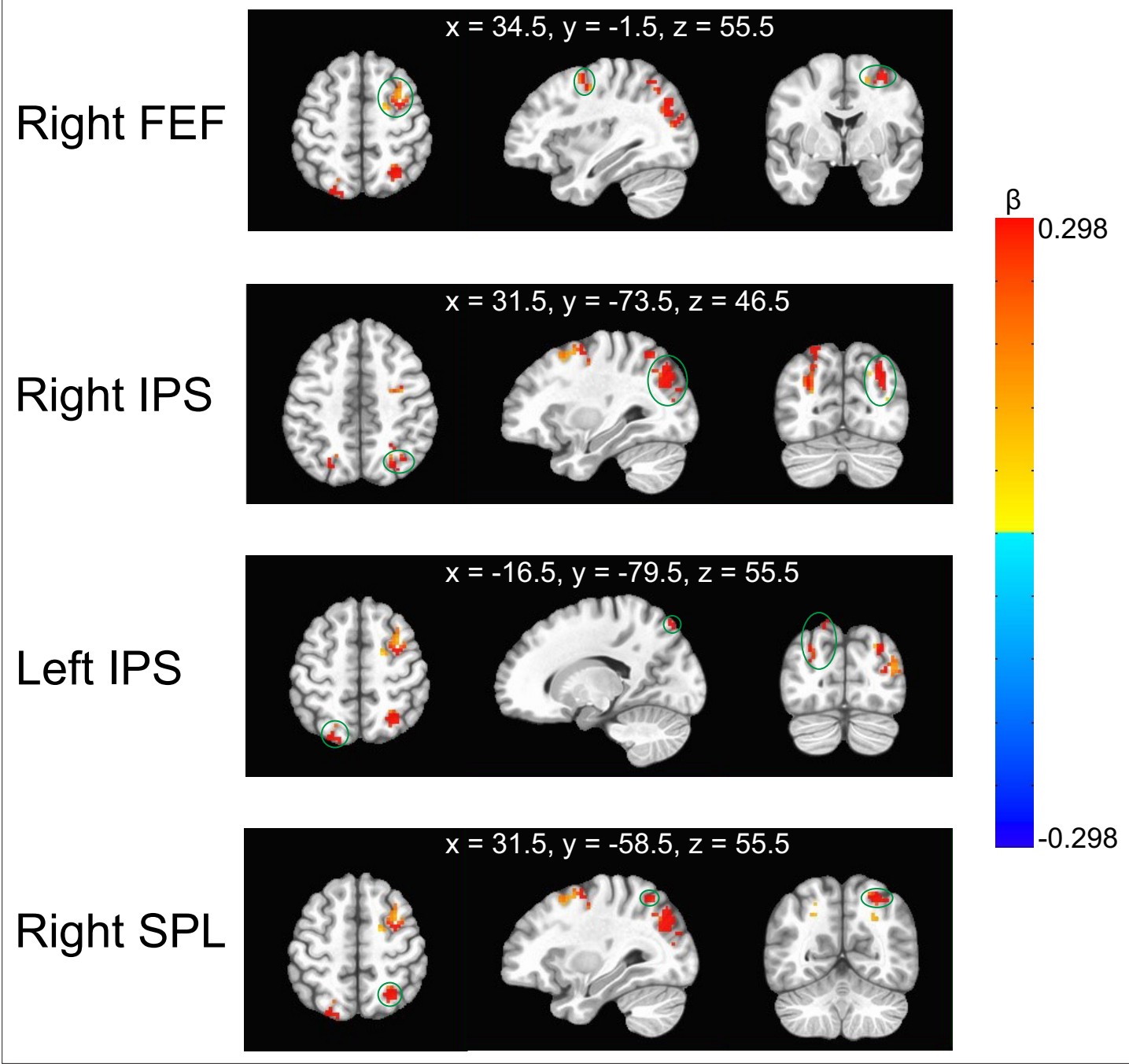

**Figure 3.** Illustration of the clusters with stronger 'dichoptic-binocular' contrast in the experimental runs than in the control runs. They were located in the right frontal eye field (FEF), bilateral intraparietal sulcus (IPS), and the right superior parietal lobule (SPL). The green circles indicate the corresponding cluster. The MNI coordinates represent the locations of peak voxels of each cluster. The color bar denotes the difference of 'dichoptic-binocular' contrast (β values) between the experimental and control runs.

to eye-based attention. To seek these brain regions, we used the AFNI program '3dttest++' to access the difference of 'dichoptic-binocular' contrast between the experimental and control runs. The AFNI program 'ClustSim' was then applied for multiple comparison correction, yielding a minimum significant cluster size of 21 voxels (voxel wise $p=0.001$; cluster threshold $\alpha=0.05$). We found four clusters showing stronger responses to the dichoptic movies than to the binocular movies, especially in the experimental runs (*Figure 3*). They are located in the right FEF, bilateral IPS, and the right superior parietal lobule (SPL). In the control runs, the responses of these areas to the dichoptic and binocular

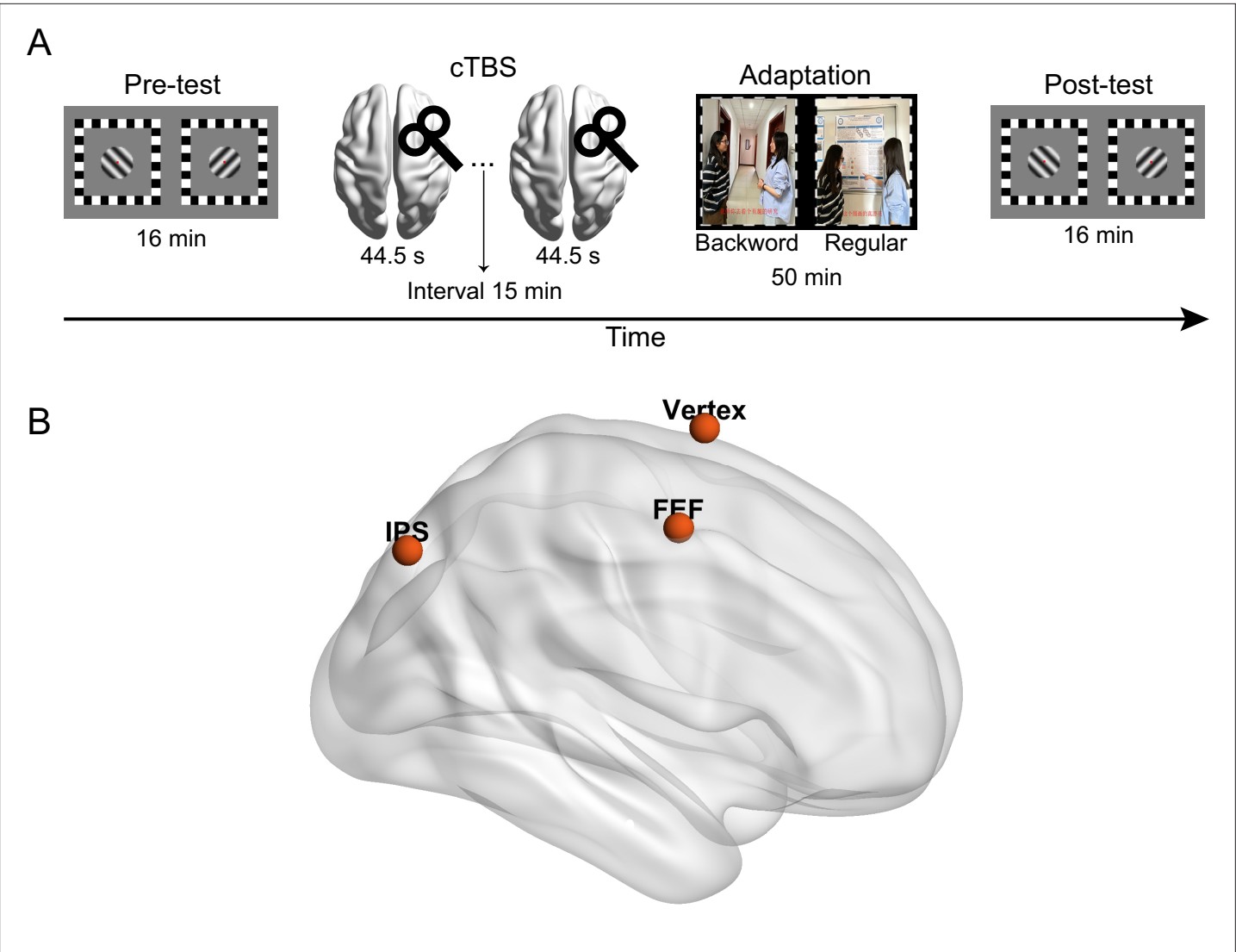

**Figure 4.** Illustration of (**A**) the process in Experiment 2 and (**B**) the stimulation sites of continuous theta burst stimulation (cTBS).

movies showed no discernible difference. As previous research has demonstrated that these brain areas are part of the dorsal attentional network (*Vossel et al., 2014*), we therefore speculate that the identified clusters would be responsible for eye-based attention.

## Experiment 2: Suppressing FEF with cTBS attenuated the ocular dominance shift induced by eye-based attention

The results of Experiment 1 suggest that areas within the fronto-parietal network are responsible for eye-based attention. To further ascertain the causal role of eye-based attention in reshaping the ocular dominance, we then measured the effect of 'dichoptic-backward-movie' adaptation (*Song et al., 2023*) on ocular dominance with the function of any identified fronto-parietal area inhibited by cTBS. If suppressing any of the fronto-parietal regions results in a reduction or disappearance of ocular dominance shift after adaptation, we can infer that eye-based attention plays a causal role in the so-called attention-induced ocular dominance plasticity (*Song et al., 2023*).

Participants completed a binocular rivalry task for the measurement of their ocular dominance before (pre-test) and after adaptation (post-test) (*Figure 4A*). cTBS was delivered over one of the areas identified in Experiment 1 (i.e. right FEF, right IPS) or a control region (vertex, *Wang et al., 2020*) after the pre-test (*Figure 4B*). Then, participants adapted to the 'dichoptic-backward-movie' in which regular movie images were presented to the dominant eye to maximize the effect of eye

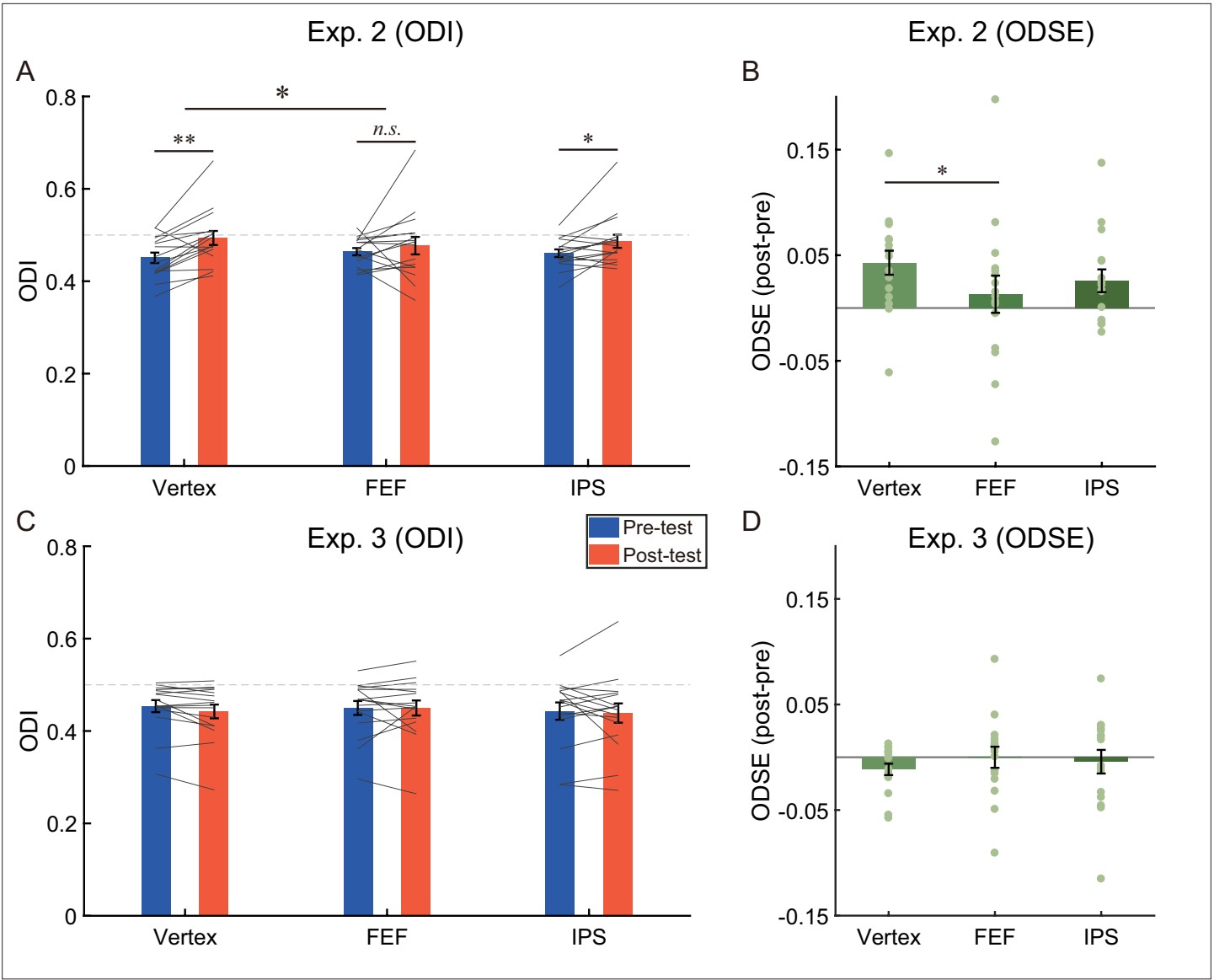

**Figure 5.** The results of (**A**) the ocular dominance index (ODI), (**B**) the ocular dominance shift effects (ODSE) in Experiment 2 (N = 16), (**C**) the ODI, and (**D**) the ODSE in Experiment 3 (N = 16). The bars show the grand average data for each condition. The individual data are plotted with gray lines or dots. The dashed gray line represents the absolute balance point for the two eyes (ODI=0.5). Error bars indicate standard errors of means. A repeated measures ANOVA was used to investigate the change of ocular dominance. Post-hoc tests were conducted using paired *t*-tests (2-tailed significance level at α = 0.05), and the resulting *p*-values were corrected for multiple comparisons using the false discovery rate (FDR) method. *p<0.05; **p<0.01; *n.s.* p>0.05.

dominance shift (*Song et al., 2023*). Meanwhile, they were asked to detect some infrequent blob targets presented in the movie images in one eye.

## Binocular rivalry test

To quantify perceptual eye dominance, we calculated the ocular dominance index (ODI) based on binocular rivalry dynamics. An increase in the value of ODI signifies a shift of ocular dominance towards the unattended eye. A 2 (test phase: pre-test, post-test)×3 (stimulation site: Vertex, FEF, IPS) repeated measures ANOVA on the ODI (*Figure 5A*) showed a significant main effect of test phase ($F_{(1,15)}$=5.12, p=0.039, $\eta^2$=0.26) but not the main effect of stimulation site ($F_{(2,30)}$=0.24, p=0.79, $\eta^2$=0.02), suggesting that the ODI in the post-test (M=0.49, SE=0.02) was greater than that in the pre-test (M=0.46, SE=0.01, 95% CI=[0.002 0.053]).

More importantly, the interaction between the test phase and the stimulation site was significant ($F$ (2,30)=3.67, p=0.038, $\eta^2$=0.20). Post-hoc test revealed that after adaptation a significant bias of ocular dominance in favor of the unattended eye could be observed if cTBS was delivered to vertex (ODI$_{pre}$=0.451 ± 0.011; ODI$_{post}$=0.494 ± 0.015; $t$ (15)=3.76, p=0.002, $d$=0.94, 95% CI=[0.019 0.067], $p_{FDR}$=0.006) and IPS (ODI$_{pre}$=0.461 ± 0.008; ODI$_{post}$=0.486 ± 0.014; $t$ (15)=2.39, p=0.030, $d$=0.60, 95% CI=[0.003 0.049], $p_{FDR}$=0.045), but not FEF (ODI$_{pre}$=0.463 ± 0.008, ODI$_{post}$=0.477 ± 0.019; $t$ (15)=0.74, p=0.47, $d$=0.19, 95% CI=[−0.025 0.051]), indicating a sharp reduction of aftereffect for stimulation at the FEF only.

We further compared the magnitude of ocular dominance shift across different stimulation sites. For this goal, we computed the ocular dominance shift effect (ODSE) by subtracting the ODI in the pre-test from that in the post-test, then compared the ODSE between the stimulation sites using paired sample $t$-tests. As shown in *Figure 5B*, the ODSE after stimulating the vertex ($M$=0.043, SE=0.011) was greater than that after stimulating the FEF ($M$=0.013, SE=0.018; $t$ (15)=2.76, p=0.015, $d$=0.69, 95% CI=[0.007 0.053], $p_{FDR}$=0.044). No significant difference was found between other pairs (vertex vs. IPS, $M$=0.026, SE=0.011; $t$ (15)=1.70, p=0.11, $d$=0.43, 95% CI=[−0.004 0.039], $p_{FDR}$=0.17; FEF vs. IPS, $t$ (15)=−1.05, p=0.31, $d$=0.26, 95% CI=[−0.038 0.013]).

## Blob detection test

For each experimental condition, the target detection rate was calculated by dividing the summed number of detected blob targets by the total number of blob targets. A 2 (eye: attended eye, unattended eye)×3 (stimulation site: Vertex, FEF, IPS) repeated measures ANOVA on the detection performance showed a significant main effect of eye ($F$ (1,15)=112.65, p<0.001, $\eta^2$=0.88) but no significant main effect of stimulation site ($F$ (2,30)=1.79, p=0.18, $\eta^2$=0.11) or the interaction ($F$ (2,30)=0.26, p=0.78, $\eta^2$=0.02), suggesting that the overall detection performance for the attended eye ($M$=0.89, SE=0.04) was better than that for the unattended eye ($M$=0.23, SE=0.06, 95% CI=[0.53 0.79]). These results confirm that the interocular difference of detection rate unaffected by the cTBS.

## Experiment 3: cTBS affected ocular dominance plasticity rather than the task performance reflecting ocular dominance

The results of Experiment 2 support the notion that eye-based attention was the cause of attention-induced ocular dominance plasticity. However, an alternative account is that the significant two-way interaction between test phase and stimulation site did not stem from any persistent malfunction of FEF in modulating ocular dominance, but rather it was due to some abnormality of binocular rivalry measures in the post-test that occurred after stimulation at the FEF only (and not at the other two brain sites). For instance, stimulation at the FEF might simply reduce the ODI measured in the binocular rivalry post-test.

Therefore, we conducted Experiment 3 to examine how suppression of the three target sites would impact binocular rivalry performance, in case that any unknown confounding factors, which were unrelated to adaptation but related to binocular rivalry measures, contributed to the results. The task procedure was similar to that in Experiment 2 except that participants watched identical regular movie images with both eyes and were no longer required to perform the blob detection task. Since there was no dichoptic-backward-movie adaptation in this experiment, we expected no shift of ocular dominance at least for stimulating the vertex.

A 2 (test phase: pre-test, post-test)×3 (stimulation site: Vertex, FEF, IPS) repeated measures ANOVA on the ODI showed neither a significant main effect nor the interaction (*Figure 5C*, $p$s >0.43). Moreover, in order to further examine these null effects, we conducted the Bayesian repeated measures ANOVA using JASP with default priors and computed inclusion Bayes factors ($BF_{incl}$) which suggest the evidence for the inclusion of a particular effect calculated across matched models. Specifically, the Bayesian ANOVA provided moderate evidence ($BF_{incl}$=0.28) to support the null hypothesis of no main effect of the stimulation site and weak evidence ($BF_{incl}$=0.39) in support of the null hypothesis of no main effect of test phase. Additionally, the Bayesian ANOVA yielded moderate evidence ($BF_{incl}$=0.23) supporting the null hypothesis of no interaction effect between the test phase and the stimulation site. These results suggested that the suppression of the fronto-parietal cortex had no impact on performance in binocular rivalry tasks.

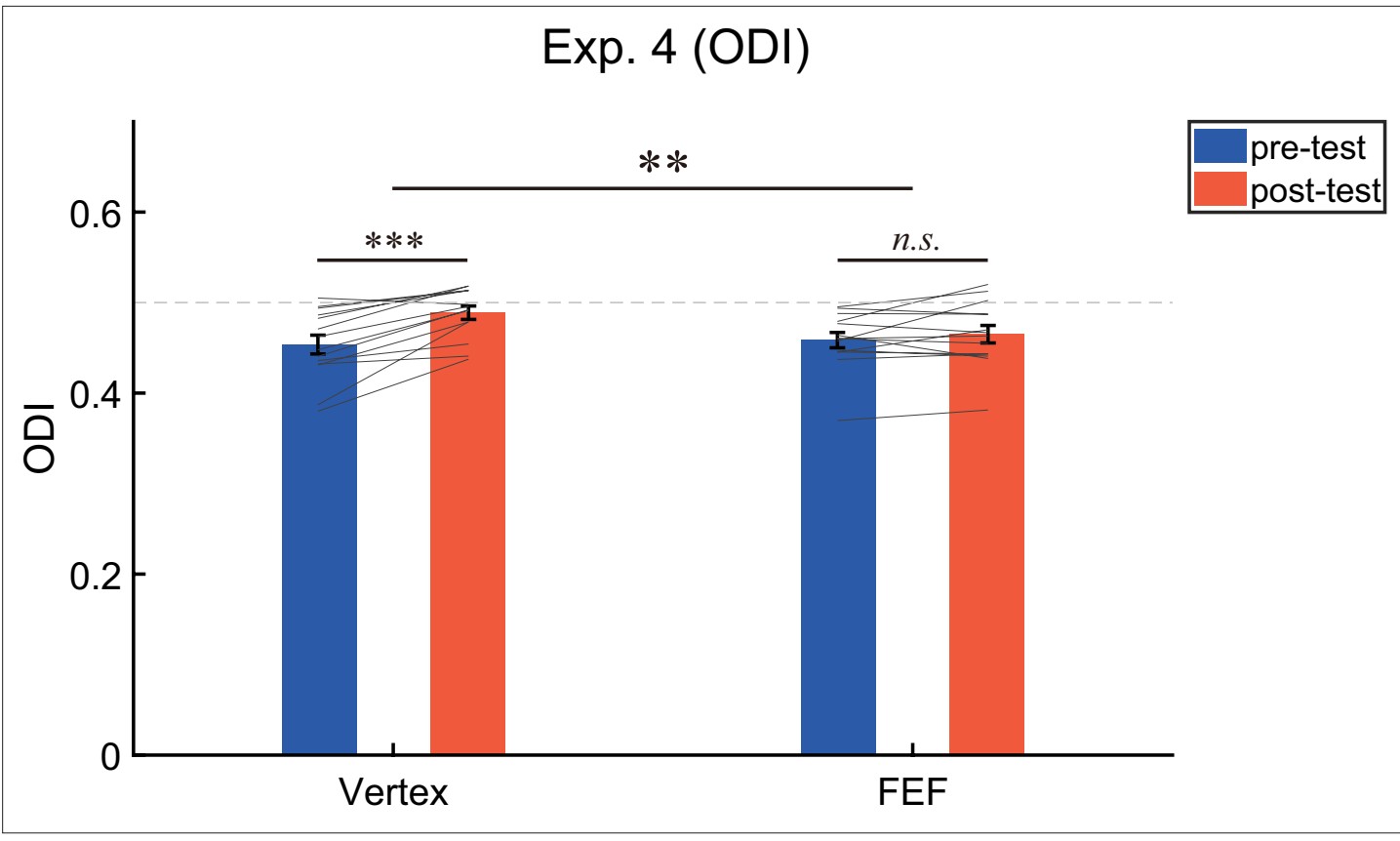

**Figure 6.** The results of the ocular dominance index (ODI) in Experiment 4 (N = 14). The bars show the grand average data for each condition. The individual data are plotted with gray lines. The dashed gray line represents the absolute balance point for the two eyes (ODI=0.5). Error bars indicate standard errors of means. A repeated measures ANOVA was used to investigate the change of ocular dominance. Post-hoc tests were conducted using paired *t*-tests (2-tailed significance level at α = 0.05), and the resulting *p*-values were corrected for multiple comparisons using the FDR method. **p<0.01; ***p<0.001; *n.s.* p>0.05.

### Experiment 4: Sound elimination did not impair the blob detection performance following the suppression of FEF

The suppression of FEF via cTBS in Experiment 2 did not impair the performance of blob detection for the attended eye. This result seems confusing, as the interocular difference of blob detection rate is presumed to be a coarse estimation of the real-time effect of eye-based attention. One possible explanation is that with the help of synchronized audio in Experiment 2, residual attentional control following the cTBS might still be competent to sustain attention to the regular movie, allowing a sound performance for this relatively easy task (detection targets appeared only once every 5 min in each eye). To examine this explanation, we conducted Experiment 4, wherein the task procedure was similar to that in Experiment 2 except that sound was no longer presented during the dichoptic-backward-movie adaptation and cTBS was only applied to FEF and vertex. If the synchronized audio (with the regular movie) really contributes to the performance of blob detection for the attended eye after the FEF was suppressed, eliminating the sound may lead to close performance for the attended and unattended eyes.

### Binocular rivalry test

A 2 (test phase: pre-test, post-test)×2 (stimulation site: Vertex, FEF) repeated measures ANOVA on the ODI (*Figure 6*) showed a significant main effect of test phase ($F_{(1,13)}$=22.94, p<0.001, $\eta^2$=0.64) but not the main effect of stimulation site ($F_{(1,13)}$=2.67, p=0.13, $\eta^2$=0.17), suggesting that the ODI in the post-test (M=0.48, SE=0.01) was greater than that in the pre-test (M=0.46, SE=0.01, 95% CI=[0.01 0.03]). More importantly, the interaction between the test phase and the stimulation site was significant ($F_{(1,13)}$=12.76, p=0.003, $\eta^2$=0.50). Post-hoc test revealed that after adaptation a significant

bias of ocular dominance in favor of the unattended eye could be observed if cTBS was delivered to vertex ($ODI_{pre}$=0.454 ± 0.01; $ODI_{post}$=0.489 ± 0.007; $t$ (13)=5.41, p<0.001, $d$=1.45, 95% CI=[0.021 0.049], $p_{FDR}$ <0.001), but not FEF ($ODI_{pre}$=0.459 ± 0.008, $ODI_{post}$=0.465 ± 0.01; $t$ (13)=1.24, p=0.24, $d$=0.33, 95% CI=[–0.005 0.018]).

## Blob detection test

A 2 (eye: attended eye, unattended eye)×2 (stimulation site: Vertex, FEF) repeated measures ANOVA on the target detection rate showed a significant main effect of an eye ($F$ (1,13)=101.01, p<0.001, $\eta^2$=0.89) but no significant main effect of stimulation site ($F$ (1,13)=1.60, p=0.23, $\eta^2$=0.11) or the interaction ($F$ (1,13)=0.10, p=0.76, $\eta^2$=0.01), suggesting that the overall detection performance for the attended eye ($M$=0.79, SE=0.04) was better than that for the unattended eye ($M$=0.23, SE=0.05, 95% CI=[0.44 0.68]). These results indicate that the interocular difference in detection rate remains unaffected by the cTBS even if no sound was presented.

## Discussion

The present study investigated the neural mechanisms underlying the shift of ocular dominance induced by 'dichoptic-backward-movie' adaptation. By using fMRI-guided TMS, we provide direct causal evidence for the effect of eye-based attention in modulating ocular dominance, and disclose the crucial role of FEF.

To our knowledge, our Experiment 1 for the first time explored the neural mechanisms of top-down eye-based attention. As compared to when participants watched a movie binocularly, we found stronger activation in FEF, IPS, and SPL when they watched the same movie presented in one eye while ignoring the backward movie in the other eye. Since focusing on the movie in one eye necessitates more attention allocated to stimuli in that eye, the activation of these fronto-parietal areas likely reflects their functions in controlling eye-based attention. Given that the activated areas belong to the dorsal attentional network (*Vossel et al., 2014*), we advocate that top-down eye-based attention is also controlled by the dorsal attentional network. Furthermore, the activation for eye-based attention was predominantly localized in the right hemisphere, showing a typical right-hemisphere functional dominance of attentional control (*Duecker et al., 2013*; *Mayrhofer et al., 2019*; *Sack, 2010*).

Studies on the real-time effect of eye-based attention have shown that top-down eye-based attention can promote the attended eye's dominance (*Wong et al., 2021*; *Zhang et al., 2012*). Yet both recent (*Song et al., 2023*; *Wang et al., 2021*) and the present work has suggested that prolonged eye-based attention can lead to a counterintuitive perceptual aftereffect, a shift of ocular dominance towards the unattended eye. Interestingly, in our Experiment 2 this aftereffect was significantly attenuated after we temporarily inhibited the cortical function of FEF via cTBS. This finding indicates the crucial role of FEF in the formation of attention-induced ocular dominance shifts.

To exclude the possibility that the cTBS on FEF simply affected the binocular rivalry measures, we conducted Experiment 3, the procedure of which resembled that of Experiment 2 except that participants were presented with regular movie images binocularly. This modification eliminated any involvement of eye-based attentional allocation. The results indicated that the cTBS delivered to FEF did not produce any significant changes in the binocular rivalry performance, thus supporting the notion that inhibition of the FEF undermined its modulatory role in ocular dominance plasticity during adaptation, which, in turn, caused a substantial reduction of ocular dominance shift towards the unattended eye.

Although both the FEF and IPS were found to be responsible for top-down eye-based attention in Experiment 1, the delivery of cTBS to IPS did not affect the adaptation-induced ocular dominance shift. Indeed, both areas belong to the dorsal attentional network (*Vossel et al., 2014*), yet previous research has shown some functional distinctions between the two areas. For instance, it is found that the FEF plays a pivotal role in filtering out distractors (*Lega et al., 2019*) and sustaining attention (*Esterman et al., 2015*), whereas no such effect has been observed in the IPS. Furthermore, stimulation of the FEF (*Ruff et al., 2006*; *Veniero et al., 2021*) and IPS (*Ruff et al., 2008*) can both modulate activities in early visual areas, but produce qualitatively different effects. In this vein, our work reveals a new type of functional distinction between the two areas, showing a unique role of the FEF in attention-induced ocular dominance plasticity.

Then how does FEF regulate the attention-induced ocular dominance shift? Our previous work has found that the aftereffect (for simplicity, hereafter we use aftereffect to denote the attention-induced ocular dominance shift) can be produced only when the adapting stimuli involve adequate interocular competition, and is measurable only when the testing stimuli are not binocularly fused (*Song et al., 2023*). Given the indispensability of interocular competition, we explained those findings in the framework of the ocular-opponency-neuron model of binocular rivalry (*Said and Heeger, 2013*). The model suggests that there are some opponency neurons which receive excitatory inputs from monocular neurons for one eye and inhibitory inputs from monocular neurons for the other eye (e.g. AE-UAE opponency neurons receive excitatory inputs from the attended eye (AE) and inhibitory inputs from the unattended eye (UAE)). Then a difference signal is computed so that the opponency neurons fire if the excitatory inputs surpass the inhibitory inputs. Upon activation, the opponency neurons will in turn suppress the monocular neurons which send inhibitory signals to them.

Based on this model, we proposed an ocular-opponency-neuron adaptation account to explain the aftereffect, and pointed out that the attentional system likely modulated the AE-UAE ocular opponency neurons (*Song et al., 2023*). So why would FEF modulate the AE-UAE opponency neurons? The reason may be twofold. Firstly, understanding the logic during the dichoptic-backward-movie viewing may require filtering out the distracting information (from the unattended eye) and sustaining attention (to the attended eye), which is exactly the role of FEF (*Esterman et al., 2015*; *Lega et al., 2019*). Secondly, due to the special characteristics of binocular vision system, filtering the distracting input from the unattended eye may have to rely on the interocular suppression mechanism. According to the ocular-opponency-neuron model, this is achieved by the firing of the AE-UAE opponency neurons that send inhibitory signals to the UAE monocular neurons.

As mentioned previously, the firing of the AE-UAE opponency neurons requires stronger activity for the AE monocular neurons than for the UAE monocular neurons. This is confirmed by the results shown in Figure 8 of *Song et al., 2023* that the monocular response for the attended eye during the entire adaptation phase was slightly stronger than that for the unattended eye. Accordingly, during adaptation the AE-UAE opponency neurons were able to activate for a longer period and thus adapted to a larger extent than the UAE-AE opponency neurons. This would cause the monocular neurons for the unattended eye to receive less inhibition from the AE-UAE opponency neurons in the post-test as compared with the pre-test, leading to a shift of ocular dominance towards the unattended eye. In this vein, the magnitude of this aftereffect should be proportional to the extent of adaptation of the AE-UAE relative to UAE-AE opponency neurons. Attentional enhancement on the AE-UAE opponency neurons is believed to strengthen this aftereffect, as it has been found that attention can enhance adaptation (*Dong et al., 2016*; *Rezec et al., 2004*). Inhibition of FEF likely led such attentional modulation to be much less effective. Consequently, the AE-UAE opponency neurons might not have the chance to adapt to a sufficiently larger extent than the UAE-AE opponency neurons, leading to a statistically non-detectable aftereffect in Experiment 2. Therefore, the results of Experiments 2–4 in the present study suggest that within the context of the ocular-opponency-neuron adaptation account, FEF might be the core area to fulfill the attentional modulations on the AE-UAE opponency neurons.

An unresolved issue is why inhibiting the cortical function of FEF did not impair the performance of the blob detection task. One potential explanation is that the synchronized audio in Experiment 2 might help increase the length of time that the regular movie dominated awareness. However, the results of Experiment 4 did not support this explanation, in which the performance of blob detection survived from the inhibition of FEF even when silent movies were presented. Although this issue remains to be explored in future work, it does not contradict with our notion of FEF modulating AE-UAE opponency neurons. It should be noted that our notion merely states that FEF is the core area for attentional modulations on activities of AE-UAE opponency neurons. No other role of FEF during the adaptation is assumed here (e.g. boosting monocular responses or increasing the conscious level of stimuli in the attended eye). In contrast, according to the most original definition, the blob detection performance serves as an estimation of visibility (or consciousness level) of the stimuli input from each eye, despite the initial goal of adopting this task is to precisely quantify eye-based attention (which might be impractical). Thus, according to our notion, inhibition of FEF does not necessarily lead to the deteriorate performance of blob detection. Furthermore, our findings consistently indicated that the visibility of stimuli in the attended eye was markedly superior

to that of stimuli in the unattended eye, yet the discrepancy in the SSVEP monocular responses between the two eyes was minimal though it had reached statistical significance (*Song et al., 2023*). Therefore, blob detection performance in our work may only faithfully reflect the conscious level in each monocular pathway, but it is probably not an appropriate index tightly associated with the attentional modulations on monocular responses in early visual areas. Indeed, previous work has argued that attention but not awareness modulates neural activities in V1 during interocular competition (*Watanabe et al., 2011*), but see *Yuval-Greenberg and Heeger, 2013*. We have noticed and discussed the counterintuitive results of blob detection performance in our previous work (*Song et al., 2023*). Here, with the new counterintuitive finding that inhibition of FEF did not impair the performance of blob detection, we suspect that blob detection performance in the 'dichoptic-backward-movie' adaptation paradigm may not be an ideal index that can be used to accurately quantify eye-based attention.

In summary, the present study for the first time discovers that the fronto-parietal attentional network is involved in controlling eye-based attention in the 'dichoptic-backward-movie' adaptation paradigm. Moreover, the present findings provide direct causal evidence for the role of eye-based attention in modulating ocular dominance. This causal role of eye-based attention is distinct from the homeostatic compensation mechanism commonly used to explain the effect of short-term monocular deprivation (*Bai et al., 2017*; *Lunghi et al., 2011*; *Lunghi et al., 2015*; *Lyu et al., 2020*; *Zhou et al., 2015*; *Zhou et al., 2013*).

## Materials and methods
### Experiment 1
#### Participants
A total of 20 participants were recruited to take part in the screening process (see Stimuli and Procedure), 17 successfully passed the screening test and completed an fMRI experiment. Due to excessive head motion during fMRI scanning, the data of one participant was excluded from further analysis. Eventually, data from 16 participants (12 females, aged from 20 to 28 years) were used. The sample size was predetermined based on the previous study (*Song et al., 2023*). All participants were naive to the experimental hypotheses. They possessed normal or corrected-to-normal visual acuity, and provided informed consent in accordance with the Declaration of Helsinki. All the experiments in this study were approved (H21058, 11/01/2021) by the Institutional Review Board of the Institute of Psychology, Chinese Academy of Sciences.

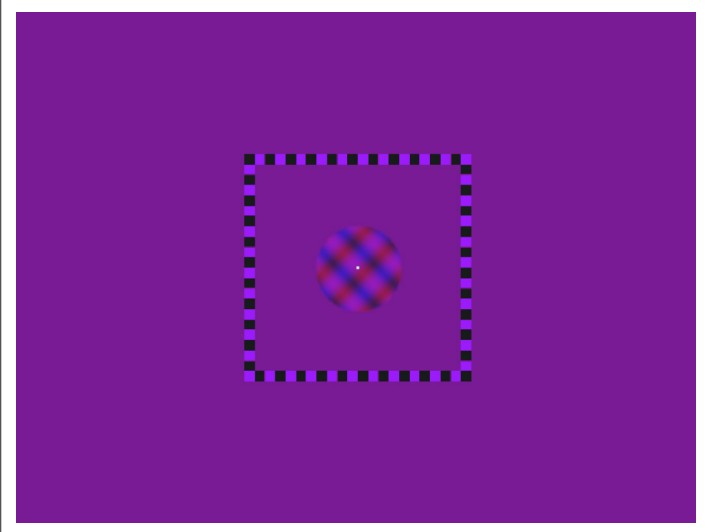

**Figure 7.** Illustration of binocular rivalry stimuli in the screening process.

## Apparatus

During the screening process, stimuli were presented on a gamma-corrected 27-inch ASUS VG279QM LCD monitor (Asus, Shanghai, China, 1920 × 1080 pixels' resolution, 60 Hz refresh rate) with the mean luminance of 30.18 cd/m². Participants viewed the stimuli through a paper red-blue stereoscopic glasses at a distance of 100 cm. A chinrest was used to stabilize the head position. The experimental stimuli were programmed using MATLAB (The MathWorks, Natick, USA) and Psychtoolbox (*Brainard, 1997*; *Pelli, 1997*). A binocular rivalry task was adopted to screen participants for extreme ocular dominance.

During the fMRI scanning, the stimuli were projected onto a flat panel screen (49 × 36.75 cm, 1024 × 768 pixels' resolution, 60 Hz refresh rate). The screen was gamma-corrected and placed in the front of the MR scanner using an LCD projector. Participants viewed the screen at the distance of 93 cm via the paper red-blue stereoscopic glasses. The experiment was conducted in a dimly lit room.

## Stimuli and procedure

### Screen phase

The stimuli were two orthogonal sinusoidal gratings (±45° from vertical, 2° in diameter, 1.5 cpd) over-lapped at the center of screen. We carefully adjusted the contrast of two gratings, so that when participants wore the red-blue glasses, they could only see the red grating through the red glass and the blue grating through the blue glass. The contrasts of red and blue gratings were 48% and 80%, respectively. In addition, a high-contrast checkerboard 'frame' (size: 5°×5°; thick: 0.25°) and a central white fixation point (diameter: 0.04°) were presented to facilitate binocular fusion (*Figure 7*).

The binocular rivalry test consisted of sixteen 1 min trials. Each trial began with a 5 s blank interval, followed by the presentation of the rival gratings for 55 s. Participants were instructed to hold down one of the three keys (Right, Left, or Down Arrow) to report the grating they perceived (clockwise, counterclockwise, or mixed). The orientation of grating presented to each eye was constant within a trial, but was counter-balanced across trials.

### fMRI scanning phase

During fMRI scanning, participants viewed movie images either dichoptically or binocularly with the red-blue glasses. In the dichoptic condition, the movie stimuli were two overlapped red and blue movie episodes (*Figure 2A*). One was played normally and the other was the same movie but played in a backward sequence (*Song et al., 2023*). The regular movie images were presented to the domi-nant eye and the backward ones to the non-dominant eye (for the definition of eye dominance, see Experimental Design below). In the binocular condition, the stimuli were regular movie images colored in purple to ensure visible inputs from both eyes (*Figure 2B*).

To create the colored movie stimuli, the original movie images were first transformed into gray-scale. In theory, the red (or blue) movie images can then be generated by zeroing the values of Green and Blue (or Red and Green) channels. However, empirically, the contrast of the red movie images was reduced by applying a multiplication factor of 0.6 to the values of the Red channel and setting the value of the Blue channel to 80 to achieve perfect monocular stimulation in each eye. The red movie images and the blue movie images were then overlapped to generate the red-blue movie images used in the dichoptic condition (*Figure 2D*). The purple movie images were obtained by first gener-ating the same red and blue regular movie images in a similar methodology as that employed for the aforementioned red-blue movie images, followed by their overlap. To facilitate binocular fusion, a gray fixation point (diameter: 0.33°) positioned at the center of the movie images and a high-contrast checkerboard 'frame' (size: 17.73° × 23.85°; thick: 0.27°) surrounding the movie were presented during stimulation. The frame rate of movies was 30 fps.

A block design was adopted to study the neural mechanisms related to eye-based attention. A movie block lasted for 30 s, alternated with 12–18 s (12, 14, 16, or 18 s) blanks. Each run consisted of eight movie blocks. The stimuli of each block could be either the dichoptic movie or the binocular movie, with the sequence counter-balanced.

The fMRI experiment included experimental runs and control runs. During an experimental run, participants were instructed to attentively watch the regular movie and comprehend the plot. To facil-itate participants' understanding of the plot, the sound track for the movie was played synchronized with the regular movie images. After the scanning of each run, a question on the movie plot would be

presented on the screen. Participants needed to answer the question via button press, allowing us to examine if they had well followed the plot during the scanning. In a control run, participants performed a rapid serial visual presentation (RSVP) task while watching the movie stimuli (*Figure 2C*). A series of capital letters were presented to both eyes on the center of the screen. Each letter subtended 0.23° × 0.27°, and was presented for 153 ms. Participants' task was to press the button whenever they found an 'X' in the letter stream. Within each block, the letter 'X' appeared for a total of three times. Participants received a feedback about their task performance after the completion of each run. To ensure that participants focused their attention on the RSVP task, the soundtracks were not played during the control runs.

## Experimental design

In the screening test, participants performed three sessions of binocular rivalry tests to measure perceptual ocular dominance. In the binocular rivalry tests, at any given time only the grating in one eye reached awareness and remained visible for a while before the grating in the other eye competed for awareness. The dominant eye was defined as the one that showed longer summed phase durations. Because perceptual ocular dominance fluctuated greatly in the first few trials of a day (*Suzuki and Grabowecky, 2007*), five warm-up trials were completed before the formal task session, the data of which were not analyzed. Perceptual ocular dominance in each test was evaluated by computing an ODI with the formula $T_L / (T_L + T_R)$, where $T_L$ and $T_R$ represented the summed phase durations for perceiving the stimulus in the left eye and right eye, respectively. Extreme ocular dominance can potentially impede accurate measures of attentional effects. For instance, if the dominant eye is excessively strong, participants may effortlessly perceive the regular movie images for an extended duration, which makes voluntary eye-based attention dispensable. Therefore, participants with ODI values greater than 0.67 or less than 0.33 (3 participants in total) were not allowed to participate in the fMRI scanning.

Each participant completed 3 experimental runs and three control runs, with the order counterbalanced across participants. Each run consisted of the movie blocks of dichoptic and binocular conditions.

## fMRI data acquisition and preprocessing

fMRI data were collected by a Siemens 3T Magnetom Trio scanner with a 20-channel phased array head coil. T1 weighted anatomical images were acquired at the beginning of scanning (repetition time (TR)=2600 ms, echo time (TE)=3.02 ms, flip angle (FA)=8°, field of view (FOV)=256 mm, slices = 176, voxel size=1.0 mm × 1.0 mm × 1.0 mm). Functional data were acquired with T2* weighted Echo-planar imaging (EPI) sequence (TR=2000 ms, TE=30ms, FA=90°, FOV=200 mm, slices=32, voxel size=3.1 mm × 3.1 mm × 3.5 mm).

The first three volumes were discarded for magnetization equilibrium. The fMRI data were preprocessed using Analysis of Functional NeuroImages (AFNI) software (*Cox, 1996*). The preprocessing steps encompassed slice timing correction, motion correction, aligning the functional image with anatomy, spatial normalization (MNI152 space), spatial smoothing with 4 mm Gaussian kernel (full width at half maximum, FWHM), and scaling each voxel time series to have a mean of 100. Data from one participant was excluded from further analysis due to the head motion of more than 10% of volumes exceeding 0.3 mm (*Silvers et al., 2017*; *Somerville et al., 2013*).

## fMRI data analysis

We performed a single-subject general linear model (GLM) using the AFNI program '3dDeconvolve.' To create regressors for each experimental conditions (dichoptic and binocular), the boxcar function representing the duration of each block (30 s) was convolved with the canonical hemodynamic response function (HRF). Volumes were censored if the motion derivatives exceeded 0.3 mm. Highpass filtering and three motion parameters were modeled as non-interest regressors to account for drift and residual motion effects. In addition, single-subject contrasts between experiment conditions (dichoptic >binocular) were calculated.

Subsequently, we used the AFNI program '3dttest++' to access the difference of the 'dichopticbinocular' contrast between the experimental and control runs. To constrict the analysis on positive activations to movie stimuli, a mask was created specifically for the t-test analysis. This mask was

generated by taking the union of all positively activated voxels across all participants for each experiment conditions. 'ClustSim' was then applied for multiple comparison correction, yielding a minimum cluster size of 21 voxels (voxelwise $p=0.001$; cluster threshold $\alpha=0.05$).

## Experiment 2

### Participants

Twenty-one participants were initially recruited to participate in the experiment. Eighteen of them successfully passed the screening and completed the formal experiment. The specific screening criteria can be found in the Experimental Design and Data Analysis section. Due to the failure of binocular fusion and the lack of concentration during the experiment, the data of two participants were excluded from the analysis. Accordingly, the data from 16 participants (12 females, aged from 18 to 28 years) were analyzed, among which five participants have participated in Experiment 1.

### Apparatus

Visual stimuli of the behavioral task were displayed on a gamma-corrected 27-inch ASUS VG279QM LCD monitor with the mean luminance of 29.64 cd/m². The spatial resolution of the monitor was 1920 × 1080 pixels, and the refresh rate was 60 Hz. Participants viewed visual stimuli through a mirror stereoscope with their eyes 100 cm away from the monitors. To stabilize their head position and minimize movement, participants were instructed to rest their heads on a chinrest. The behavioral tasks were performed in a dimly lit room.

## Stimuli and procedure

### Binocular rivalry test

The binocular rivalry stimuli were two achromatic sine-wave grating disks that were oriented orthogonally (±45° from vertical, diameter: 1°, spatial frequency: 3 cpd, 80% Michelson contrast) and presented foveally to each eye, respectively. A central red fixation point (0.07° in diameter) and a high-contrast checkerboard 'frame' (size: 2.5° × 2.5°; thickness: 0.25°) were also presented to both eyes to facilitate stable binocular fusion (*Figure 1A*). The procedure of binocular rivalry test was consistent with that of Experiment 1.

### Dichoptic-backward-movie Adaptation

We employed the 'dichoptic-backward-movie' paradigm for the adaptation phase as utilized in the previous research (*Song et al., 2023*). During the 50 min adaptation, participants were presented with two movie images dichoptically (*Figure 1B*), with one eye viewing the regular movie images while the other eye viewing the corresponding backward movie images. Note that the backward movie was formally identical to the regular one, except for the absence of a coherent plot. Participants were instructed to exert their utmost effort in comprehending the logic of the regular movie, while disregarding the superimposed backward movie. Therefore, top-down selective attention was supposed to be predominantly directed towards the regular movie. The eye that received the regular movie images was referred to as the 'attended eye,' while the other eye as the 'unattended eye.' The frame rate of movies was 30 fps.

### Blob detection task

The blob detection task has been detailedly described in a prior research (*Song et al., 2023*). During the adaptation phase, participants were required to watch movies while simultaneously detecting a reduction in color saturation within a blob region that was presented exclusively to one eye (*Figure 1C*). The locations of the blobs were predetermined by the experimenter, so that they always appeared on faces in the movie. Participants were required to promptly press the SPACE key upon detecting any portion of a character's face turning gray.

To ensure that the primary focus remained on movie watching, the presentation of the blob was infrequent. To determine the timing of the presentation of the blobs, the adaptation period was divided into 20 segments, each lasting for 150 s. A blob would fade in within 0.2 s at a random moment in the middle 50 s of a segment on the movie of one eye. And after 5 s, it faded out within 0.2 s. There would be 10 blobs presented to each eye.

## TMS

We utilized the Magstim Rapid[2] stimulator equipped with a standard 70 mm figure-of-eight coil to deliver cTBS. The cTBS protocol consisted of 267 bursts, with each burst comprising three pulses at 30 Hz and repeating at 6 Hz. In total, there were 801 pulses delivered and the duration of a single cTBS train was 44 s (*Cazzoli et al., 2009*; *Chaves et al., 2012*; *Dordevic et al., 2022*; *Nyffeler et al., 2008*). To extend the duration of inhibitory effects induced by the cTBS, two cTBS trains with an interval of 15 min were applied over the same stimulation site. This stimulation protocol has been demonstrated to safely extend the duration of cTBS effects to over 2 hr (*Goldsworthy et al., 2012*; *Nyffeler et al., 2006*), thereby ensuring the sustained inhibitory effect throughout the adaptation process. The stimulation intensity was set at 80% of the individual's resting motor threshold (RMT), which was defined as the minimum machine output required to elicit a visible twitch of the left index finger for 50% of pulses.

Given that the dorsal attentional network primarily consists of the FEF and the IPS (*Corbetta and Shulman, 2002*; *Mayrhofer et al., 2019*), with a functional right-hemisphere dominance (*Duecker et al., 2013*; *Mayrhofer et al., 2019*; *Sack, 2010*), we selected the right FEF and right IPS from the four clusters identified in Experiment 1 as the target regions for cTBS (*Gallotto et al., 2022*). The cTBS stimulation sites were localized based on the peak of the group effect observed in Experiment 1 (MNI coordinates: rFEF: 34.5, −1.5, 55.5; rIPS: 31.5, −73.5, 46.5. see *Figure 4B*). In addition, we applied cTBS on the vertex (MNI coordinates: 0, 0, 75) as a control condition (*Wang et al., 2020*). The coil position was guided by the Brainsight 2 neuro-navigation software based on individual T1-weighted images.

## Experimental design

Before the formal experiment, all participants underwent a 3–7 days of training period (three tests per day with a 10 min rest in between) to ensure consistent performance on the binocular rivalry task (*Bao et al., 2018*). The task procedure of each day was similar to that of the screen test in Experiment 1. One participant withdrew from the study due to difficulties in achieving binocular fusion.

After the practice of the binocular rivalry task, participants were also examined for their capacity in allocating eye-based attention (*Neisser and Becklen, 1975*). They were asked to watch the dichoptic movie while performing the blob detection task (*Song et al., 2023*). Only those who showed superior blob detection performance for the attended eyes and subjectively reported paying more attention to the regular movie images were eligible to finish the formal experiment. All the participants who participated in this stage successfully passed the screening. The brain structural images of participants who had not previously participated in Experiment 1 were then acquired.

In the formal experiment, participants performed two binocular rivalry tests initially. The first served as a warm-up test that consisted of five trials and was not analyzed (*Bai et al., 2017*; *Bao et al., 2018*). The second comprised 16 trials that measured the perceptual ocular dominance prior to adaptation (i.e. pre-test). Following the pre-test, two cTBS trains with an interval of 15 min were applied over one stimulation site. Afterward, participants completed a 50 min dichoptic-backward-movie adaption. During the adaptation phase, the main task was to comprehend the plot of the movie. Meanwhile, participants were required to detect the infrequent blob targets. The dichoptic movie was played with sound and the audio track always synchronized with the regular movie images to facilitate the allocation of eye-based attention. As shown in the previous work (*Song et al., 2023*), presenting the regular movie images to the dominant eye can maximize the effect of eye dominance shift. Therefore, here the dominant eye was always selected as the attended eye. After the end of adaptation, a 16-trial binocular rivalry test was immediately conducted as the post-test (*Figure 4A*). The formal experiment consisted of three sessions, with cTBS delivered to only one of the three stimulation sites in each session. The order of stimulation sites was counter-balanced across participants. The sessions were conducted on different days to minimize any potential carryover effect.

## Data analysis

To quantify perceptual eye dominance, we calculated the ODI based on binocular rivalry dynamics. The ODI was derived using the following formula that yielded a value ranging from 0 (indicating complete dominance of the attended eye) to 1 (indicating complete dominance of the unattended eye).

$$ODI = \frac{T_{UAE}}{T_{UAE} + T_{AE}}$$

In the above formula, $T_{AE}$ and $T_{UAE}$ denote the cumulative phase durations of perceiving the stimulus in the attended eye and stimulus in the unattended eye, respectively. Participants with ODI values larger than 0.67 or smaller than 0.33 (2 participants in total) were not allowed to participate in the formal experiment.

Statistical analyses were performed using MATLAB. A 3 (stimulation site: Vertex, FEF, IPS)×2 (test phase: pre-test and post-test) repeated measures ANOVA was used to investigate the effect of cTBS delivery on ocular dominance shift. Moreover, for the blob detection test, the target detection rate of each experimental condition was calculated by dividing the summed number of detected blob targets by the total number of blob targets. Then, a 2 (eye: attended eye, unattended eye)×3 (stimulation site: Vertex, FEF, IPS) repeated measures ANOVA on the detection performance was performed. Post-hoc tests were conducted using paired $t$-tests (two-tailed significance level at $\alpha=0.05$), and the resulting $p$-values were corrected for multiple comparisons using the FDR method (*Benjamini and Hochberg, 1995*).

## Experiment 3
### Participants
Twenty participants were recruited to participate in the experiment, of which two were excluded for extreme interocular imbalance and another two voluntarily withdrew during the screening phase. Accordingly, 16 participants (14 females, aged from 19 to 26 years) completed all the experiments, one of which had participated in Experiment 1.

### Stimuli and procedure
The task procedure was similar to that in Experiment 2 except that participants watched identical regular movie images with both eyes and were no longer required to perform the blob detection task during adaptation.

### Data analysis
In addition to the data analysis in Experiment 2, we complemented the standard inferential approach with the Bayes factor (*van den Bergh et al., 2023*; *van Doorn et al., 2021*; *Wagenmakers et al., 2018*), which allows quantifying the relative evidence that the data provide for the alternative (H$_1$) or null hypothesis (H$_0$). We conducted the Bayesian repeated measures ANOVA using JASP with default priors and computed inclusion Bayes factors ($BF_{incl}$) which suggest the evidence for the inclusion of a particular effect calculated across matched models. A BF greater than 1 provides support for the alternative hypothesis. Specifically, a BF between 1 and 3 indicates weak evidence, a BF between 3 and 10 indicates moderate evidence, and a BF greater than 10 indicates strong evidence (*van Doorn et al., 2021*). In contrast, a BF below 1 provides evidence in favor of the null hypothesis.

## Experiment 4
### Participants
Fifteen participants were recruited to participate in the experiment. One was excluded for extreme interocular imbalance. Eventually, 14 participants (11 females, aged from 18 to 28 years) completed all the experiments.

### Stimuli and procedure
The task procedure was similar to that in Experiment 2 except that the sound was no longer presented during the dichoptic-backward-movie adaptation and cTBS was only applied to FEF and vertex.

## Acknowledgements

We greatly appreciate the excellent work of the technical support staff at the Institutional Center for Shared Technologies and Facilities of the Institute of Psychology, Chinese Academy of Sciences. This

research was supported by the Ministry of Science and Technology of China (2021ZD0203800) and the National Natural Science Foundation of China (31830037 and 31871104).

## Additional information

### Funding

| Funder | Grant reference number | Author |
|---|---|---|
| Ministry of Science and Technology of the People's Republic of China | 2021ZD0203800 | Min Bao |
| National Natural Science Foundation of China | 31830037 | Min Bao |
| National Natural Science Foundation of China | 31871104 | Min Bao |

The funders had no role in study design, data collection and interpretation, or the decision to submit the work for publication.

### Author contributions

Fangxing Song, Data curation, Software, Formal analysis, Validation, Investigation, Visualization, Methodology, Writing - original draft; Xue Dong, Formal analysis, Supervision, Validation, Visualization, Writing - review and editing; Jiaxu Zhao, Jue Wang, Xiaohui Sang, Xin He, Investigation; Min Bao, Conceptualization, Resources, Supervision, Funding acquisition, Validation, Visualization, Project administration, Writing - review and editing

### Author ORCIDs

Fangxing Song http://orcid.org/0000-0002-1502-0800
Xue Dong http://orcid.org/0000-0002-9930-4260
Xin He http://orcid.org/0000-0001-9941-8738
Min Bao http://orcid.org/0000-0002-5347-9663

### Ethics

All participants were naive to the experimental hypotheses. They possessed normal or corrected-to-normal visual acuity and provided informed consent in accordance with the Declaration of Helsinki. All the experiments in this study were approved (H21058, 11/01/2021) by the Institutional Review Board of the Institute of Psychology, Chinese Academy of Sciences.

Reviewer #1 (Public Review): https://doi.org/10.7554/eLife.93213.3.sa1
Reviewer #2 (Public Review): https://doi.org/10.7554/eLife.93213.3.sa2
Reviewer #3 (Public Review): https://doi.org/10.7554/eLife.93213.3.sa3
Author Response https://doi.org/10.7554/eLife.93213.3.sa4

## Additional files

### Supplementary files
• MDAR checklist

### Data availability

Source data and source code are available on Science Data Bank (https://doi.org/10.57760/sciencedb.psych.00219, version 2).

The following dataset was generated:

| Author(s) | Year | Dataset title | Dataset URL | Database and Identifier |
|---|---|---|---|---|
| Song F, Dong X, Zhao J, Wanf J, Sang X, He X, Bao X | 2024 | Causal Role of the Frontal Eye Field in Attention-induced Ocular Dominance Plasticity | https://doi.org/10.57760/sciencedb.psych.00219 | Science Data Bank, 10.57760/sciencedb.psych.00219 |

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
