## [Editor Report · eLife assessment]

This **important** study combines psychophysics, fMRI, and TMS to reveal a causal role of FEF in generating an attention-induced ocular dominance shift, with potential relevance for clinical applications. The evidence supporting the claims of the authors is **convincing**. The work will be of broad interest to perceptual and cognitive neuroscience.

---

## [Referee Report · Reviewer #1 (Public Review)]

Summary:

Based on a "dichoptic-background-movie" paradigm that modulates ocular dominance, the present study combines fMRI and TMS to examine the role of the frontoparietal attentional network in ocular dominance shifts. The authors claimed a causal role of FEF in generating the attention-induced ocular dominance shift.

Strengths:

A combination of fMRI, TMS, and "dichoptic-background-movie" paradigm techniques is used to reveal the causal role of the frontoparietal attentional network in ocular dominance shifts. The conclusions of this paper are well supported by data.

Weaknesses:

My previous concerns have been addressed.

---

## [Referee Report · Reviewer #2 (Public Review)]

Summary

Song et al investigate the role of the frontal eye field (FEF) and the intraparietal sulcus (IPS) in mediating the shift in ocular dominance (OD) observed after a period of dichoptic stimulation during which attention is selectively directed to one eye. This manipulation has been previously found to transiently shift OD in favor of the unattended eye, similar to the effect of short-term monocular deprivation. To this aim, the authors combine psychophysics, fMRI, and transcranial magnetic stimulation (TMS). In the first experiment, the authors determine the regions of interest (ROIs) based on the responses recorded by fMRI during either dichoptic or binocular stimulation, showing selective recruitment of the right FEF and IPS during the dichoptic condition, in line with the involvement of eye-based attention. In a second experiment, the authors investigate the causal role of these two ROIs in mediating the OD shift observed after a period of dichoptic stimulation by selectively inhibiting with TMS (using continuous theta burst stimulation, cTBS), before the adaptation period (50 min exposure to dichoptic stimulation). They show that, when cTBS is delivered on the FEF, but not the IPS or the vertex, the shift in OD induced by dichoptic stimulation is reduced, indicating a causal involvement of the FEF in mediating this form of short-term plasticity. A third control experiment rules out the possibility that TMS interferes with the OD task (binocular rivalry), rather than with the plasticity mechanisms. From this evidence, the authors conclude that the FEF is one of the areas mediating the OD shift induced by eye-selective attention.

The authors have addressed the issues that I raised during the first round of review.

While the results of the new experiment (Experiment 4), leave some unresolved isssues (addressed in the discussion section), they provide a very important replication of the main result, showing that even if the observed effect is small, it is robust.

---

## [Referee Report · Reviewer #3 (Public Review)]

Summary:

This study studied the neural mechanisms underlying the shift of ocular dominance induced by "dichoptic-backward-movie" adaptation. The study is self-consistent.

Strengths:

The experimental design is solid and progressive (relationship among three studies), and all of the raised research questions were well answered.

The logic behind the neural mechanisms is solid.

The findings regarding the cTMS (especially the position/site can be useful for future medical implications).

The updated Exp4 eliminates some concerns and thus makes the results even more solid.

---

## [Author Response]

The following is the authors’ response to the original reviews.

**eLife assessment**
This important study combines psychophysics, fMRI, and TMS to reveal a causal role of FEF in generating an attention-induced ocular dominance shift, with potential relevance for clinical applications. The evidence supporting the claims of the authors is solid, but the theoretical and mechanistic interpretation of results and experimental approaches need to be strengthened. The work will be of broad interest to perceptual and cognitive neuroscience.
**Public Reviews:**

**Reviewer #1 (Public Review):**
Summary:Based on a "dichoptic-background-movie" paradigm that modulates ocular dominance, the present study combines fMRI and TMS to examine the role of the frontoparietal attentional network in ocular dominance shifts. The authors claimed a causal role of FEF in generating the attention-induced ocular dominance shift.Strengths:A combination of fMRI, TMS, and "dichoptic-background-movie" paradigm techniques is used to reveal the causal role of the frontoparietal attentional network in ocular dominance shifts. The conclusions of this paper are mostly well supported by data.Weaknesses:(1) The relationship between eye dominance, eye-based attention shift, and cortical functions remains unclear and merits further delineation. The rationale of the experimental design related to the hemispheric asymmetry in the FEF and other regions should be clarified.

Thanks for the reviewer’s comments! We have further clarified the relationship between eye dominance shift, eye-based attention, and cortical functions in the Introduction and Discussion. In the Introduction, we introduce the modulating effects of eye-based attention on eye dominance. On one hand, eye-based attention can enhance eye dominance of the attended eye in real time (see page 3 first paragraph or below):

”For instance, presenting top-down attentional cues to one eye can intensify the competition strength of input signals in the attended eye during binocular rivalry (Choe & Kim, 2022; Zhang et al., 2012) and shift the eye balance towards the attended eye (Wong et al., 2021).”

On the other hand, prolonged eye-based attention can induce a shift of eye dominance to the unattended eye (see page 3 second paragraph or below):

“In Song et al. (2023)’s “dichoptic-backward-movie” adaptation paradigm (see Figure 1B), participants are presented with regular movie images in one eye (i.e., attended eye) while the other eye (i.e., unattended eye) received the backward movie images of the same episode. They were also instructed to try their best to follow the logic of the regular movie and ignore the superimposed backward movie. Therefore, the goal-directed eye-based attention was predominantly focused on the attended eye. Song et al. (2023) found that the predominance of the unattended eye in binocular rivalry increased after one hour of adaptation to the “dichoptic-backward-movie”, indicating a shift of perceptual ocular dominance towards the unattended eye. Since the overall energy of visual input from the two eyes was balanced throughout the adaptation period, the change of ocular dominance after adaptation is thought to result from unbalanced eye-based attention rather than unbalanced input energy as in typical short-term monocular deprivation (Bai et al., 2017; Lunghi et al., 2011; Zhou et al., 2014).”

Moreover, we discussed how FEF regulates attention-induced ocular dominance shift (see page 21 second paragraph to page 23 first paragraph or below, which also respond to this reviewer’s comment of Weakness #2):

“Then how does FEF regulate the attention-induced ocular dominance shift? Our previous work has found that the aftereffect (for simplicity, hereafter we use aftereffect to denote the attention-induced ocular dominance shift) can be produced only when the adapting stimuli involve adequate interocular competition, and is measurable only when the testing stimuli are not binocularly fused (Song et al., 2023). Given the indispensability of interocular competition, we explained those findings in the framework of the ocular-opponency-neuron model of binocular rivalry (Said & Heeger, 2013). The model suggests that there are some opponency neurons which receive excitatory inputs from monocular neurons for one eye and inhibitory inputs from monocular neurons for the other eye (e.g. AE-UAE opponency neurons receive excitatory inputs from the attended eye (AE) and inhibitory inputs from the unattended eye (UAE)). Then a difference signal is computed so that the opponency neurons fire if the excitatory inputs surpass the inhibitory inputs. Upon activation, the opponency neurons will in turn suppress the monocular neurons which send inhibitory signals to them.

Based on this model, we proposed an ocular-opponency-neuron adaptation account to explain the aftereffect, and pointed out that the attentional system likely modulated the AE-UAE ocular opponency neurons (Song et al., 2023). So why would FEF modulate the AE-UAE opponency neurons? The reason may be two fold. Firstly, understanding the logic during the dichoptic-backward-movie viewing may require filtering out the distracting information (from the unattended eye) and sustaining attention (to the attended eye), which is exactly the role of FEF (Esterman et al., 2015; Lega et al., 2019).

Secondly, due to the special characteristics of binocular vision system, filtering the distracting input from the unattended eye may have to rely on the interocular suppression mechanism. According to the ocular-opponency-neuron model, this is achieved by the firing of the AE-UAE opponency neurons that send inhibitory signals to the UAE monocular neurons.

As mentioned previously, the firing of the AE-UAE opponency neurons requires stronger activity for the AE monocular neurons than for the UAE monocular neurons. This is confirmed by the results shown in Figure 8 of Song et al. (2023) that monocular response for the attended eye during the entire adaptation phase was slightly stronger than that for the unattended eye. Accordingly, during adaptation the AE-UAE opponency neurons were able to activate for a longer period thus adapted to a larger extent than the UAE-AE opponency neurons. This would cause the monocular neurons for the unattended eye to receive less inhibition from the AE-UAE opponency neurons in the post-test as compared with the pre-test, leading to a shift of ocular dominance towards the unattended eye. In this vein, the magnitude of this aftereffect should be proportional to the extent of adaptation of the AE-UAE relative to UAE-AE opponency neurons. Attentional enhancement on the AE-UAE opponency neurons is believed to strengthen this aftereffect, as it has been found that attention can enhance adaptation (Dong et al., 2016; Rezec et al., 2004). Inhibition of FEF likely led such attentional modulation to be much less effective. Consequently, the AE-UAE opponency neurons might not have the chance to adapt to a sufficiently larger extent than the UAE-AE opponency neurons, leading to a statistically non-detectable aftereffect in Experiment 2. Therefore, the results of Experiments 2-4 in the present study suggest that within the context of the ocular-opponency-neuron adaptation account, FEF might be the core area to fulfill the attentional modulations on the AE-UAE opponency neurons.”

We used the experimental design with hemispheric asymmetry in the FEF and other regions for two reasons. First, many studies have shown that the dorsal attentional network has a functional right-hemisphere dominance (Duecker et al., 2013; Mayrhofer et al., 2019; Sack, 2010). This was also indicated by the results of Experiment 1 (Figure 3). Second, we found that a recent research applying TMS to FEF and IPS stimulated only the right hemisphere (Gallotto et al., 2022). Therefore, we selected the right FEF and right IPS as the target regions for cTBS. In the Methods section of Experiment 2, we have elucidated the reasons for the selection of cTBS target regions (see page 35, first paragraph or below):

“Given that the dorsal attentional network primarily consists of the FEF and the IPS (Corbetta & Shulman, 2002; Mayrhofer et al., 2019), with a functional right-hemisphere dominance (Duecker et al., 2013; Mayrhofer et al., 2019; Sack, 2010), we selected the right FEF and right IPS from the four clusters identified in Experiment 1 as the target regions for cTBS (Gallotto et al., 2022).”

(2) Theoretically, how the eye-related functions in this area could be achieved, and how it interacts with the ocular representation in V1 warrant further clarification.

Thanks for the reviewer’s comment! In the revised manuscript, we have discussed how FEF regulates attention-induced ocular dominance shift (see page 21 second paragraph to page 23 first paragraph or the quoted paragraphs under this reviewer’s first Public comment).

**Reviewer #2 (Public Review):**
SummarySong et al investigate the role of the frontal eye field (FEF) and the intraparietal sulcus (IPS) in mediating the shift in ocular dominance (OD) observed after a period of dichoptic stimulation during which attention is selectively directed to one eye. This manipulation has been previously found to transiently shift OD in favor of the unattended eye, similar to the effect of short-term monocular deprivation. To this aim, the authors combine psychophysics, fMRI, and transcranial magnetic stimulation (TMS). In the first experiment, the authors determine the regions of interest (ROIs) based on the responses recorded by fMRI during either dichoptic or binocular stimulation, showing selective recruitment of the right FEF and IPS during the dichoptic condition, in line with the involvement of eye-based attention. In a second experiment, the authors investigate the causal role of these two ROIs in mediating the OD shift observed after a period of dichoptic stimulation by selectively inhibiting with TMS (using continuous theta burst stimulation, cTBS), before the adaptation period (50 min exposure to dichoptic stimulation). They show that, when cTBS is delivered on the FEF, but not the IPS or the vertex, the shift in OD induced by dichoptic stimulation is reduced, indicating a causal involvement of the FEF in mediating this form of short-term plasticity. A third control experiment rules out the possibility that TMS interferes with the OD task (binocular rivalry), rather than with the plasticity mechanisms. From this evidence, the authors conclude that the FEF is one of the areas mediating the OD shift induced by eye-selective attention.Strengths(1) The experimental paradigm is sound and the authors have thoroughly investigated the neural correlates of an interesting form of short-term visual plasticity combining different techniques in an intelligent way.(2) The results are solid and the appropriate controls have been performed to exclude potential confounds.(3) The results are very interesting, providing new evidence both about the neural correlates of eye-based attention and the involvement of extra-striate areas in mediating short-term OD plasticity in humans, with potential relevance for clinical applications (especially in the field of amblyopia).Weaknesses(1) Ethics: more details about the ethics need to be included in the manuscript. It is only mentioned for experiment 1 that participants "provided informed consent in accordance with the Declaration of Helsinki. This study was approved by the Institutional Review Board of the Institute of Psychology, Chinese Academy of Sciences". (Which version of the Declaration of Helsinki? The latest version requires the pre-registration of the study. The code of the approved protocol together with the code and date of the approval should be provided.) There is no mention of informed consent procedures or ethics approval for the TMS experiments. This is a huge concern, especially for brain stimulation experiments!

Response: Thanks for the reviewer’s comment! In the revised manuscript, we have provided the code of the approved protocol and date of the approval (see page 25 second paragraph or below):

“This study was approved (H21058, 11/01/2021) by the Institutional Review Board of the Institute of Psychology, Chinese Academy of Sciences.”

Indeed, ethics approval and informed consent were obtained for each experiment. To avoid duplication in the text, we only presented the ethics instructions in the Methods section of Experiment 1. We have now clarified in that section that all the experiments in this study were approved by the IRB in our Institute.

(2) Statistics: the methods section should include a sub-section describing in detail all the statistical analyses performed for the study. Moreover, in the results section, statistical details should be added to support the fMRI results. In the current version of the manuscript, the claims are not supported by statistical evidence.

Response: Thanks for the reviewer’s suggestion! In the Methods section of revised manuscript, we have added a section to describe the detailed statistical analyses for each experiment (see page 37 last paragraph for Experiment 2 and page 38 last paragraph for Experiment 3 or below):

“Statistical analyses were performed using MATLAB. A 3 (stimulation site: Vertex, FEF, IPS) × 2 (test phase: pre-test and post-test) repeated measures ANOVA was used to investigate the effect of cTBS delivery on ocular dominance shift. Moreover, for the blob detection test, the target detection rate of each experimental condition was calculated by dividing the summed number of detected blob targets by the total number of blob targets. Then, a 2 (eye: attended eye, unattended eye) × 3 (stimulation site: Vertex, FEF, IPS) repeated measures ANOVA on the detection performance was performed. Post-hoc tests were conducted using paired t-tests (2-tailed significance level at α = 0.05), and the resulting p-values were corrected for multiple comparisons using the false discovery rate (FDR) method (Benjamini & Hochberg, 1995).”

“In addition to the data analysis in Experiment 2, we complemented the standard inferential approach with the Bayes factor (van den Bergh et al., 2023; van Doorn et al., 2021; Wagenmakers et al., 2018), which allows quantifying the relative evidence that the data provide for the alternative (H1) or null hypothesis (H0). We conducted the Bayesian repeated measures ANOVA using JASP with default priors and computed inclusion Bayes factors (BFincl) which suggest the evidence for the inclusion of a particular effect calculated across matched models. A BF greater than 1 provides support for the alternative hypothesis. Specifically, a BF between 1 and 3 indicates weak evidence, a BF between 3 and 10 indicates moderate evidence, and a BF greater than 10 indicates strong evidence (van Doorn et al., 2021). In contrast, a BF below 1 provides evidence in favor of the null hypothesis.”

Furthermore, in the Results section of revised manuscript, we have added the statistical details to support the fMRI results (see page 9 last paragraph or below):

“To seek these brain regions, we used the AFNI program “3dttest++” to access the difference of ‘dichoptic-binocular’ contrast between the experimental and control runs. The AFNI program “ClustSim” was then applied for multiple comparison correction, yielding a minimum significant cluster size of 21 voxels (voxel wise p = .001; cluster threshold α = 0.05). We found 4 clusters showing stronger responses to the dichoptic movies than to the binocular movies especially in the experimental runs.”

(3) Interpretation of the results: the TMS results are very interesting and convincing regarding the involvement of the FEF in the build-up of the OD shift induced by dichoptic stimulation, however, I am not sure that the authors can claim that this effect is related to eye-based attention, as cTBS has no effect on the blob detection task during dichoptic stimulation. If the FEF were causally involved in eye-based attention, one would expect a change in performance in this task during dichoptic stimulation, perhaps a similar performance for the unattended and attended eye. The authors speculate that the sound could have an additional role in driving eye-based attention, which might explain the lack of effect for the blob discrimination task, however, this hypothesis has not been tested.

Response: Thanks for the reviewer’s comment! Following this reviewer’s insightful suggestion, we have conducted a new experiment to examine the effect of sound on blob detection task (see Experiment 4 in the revised manuscript). The procedure was similar to that of Experiment 2 except that the sound was no longer presented during the dichoptic-backward-movie adaptation. The results showed that the interocular difference of blob detection rate after sound elimination remained unaffected by the cTBS, which disagreed with our explanation in the previous version of manuscript. Based on the new data, we now question the validity to use the blob detection rate to precisely quantify eye-based attention, and have tried to explain why the blob detection results do not contradict with our account for the function role of FEF in modulating the aftereffect in the Discussion of the revised manuscript (see page 23 second paragraph to page 24 first paragraph or below):

“An unresolved issue is why inhibiting the cortical function of FEF did not impair the performance of blob detection task. One potential explanation is that the synchronized audio in Experiment 2 might help increase the length of time that the regular movie dominated awareness. However, the results of Experiment 4 did not support this explanation, in which the performance of blob detection survived from the inhibition of FEF even when silent movies were presented. Although this issue remains to be explored in future work, it does not contradict with our notion of FEF modulating AE-UAE opponency neurons. It should be noted that our notion merely states that FEF is the core area for attentional modulations on activities of AE-UAE opponency neurons. No other role of FEF during the adaptation is assumed here (e.g. boosting monocular responses or increasing conscious level of stimuli in the attended eye). In contrast, according to the most original definition, the blob detection performance serves as an estimation of visibility (or consciousness level) of the stimuli input from each eye, despite the initial goal of adopting this task is to precisely quantify eye-based attention (which might be impractical). Thus, according to our notion, inhibition of FEF does not necessarily lead to deteriorate performance of blob detection. Furthermore, our findings consistently indicated that the visibility of stimuli in the attended eye was markedly superior to that of stimuli in the unattended eye, yet the discrepancy in the SSVEP monocular responses between the two eyes was minimal though it had reached statistical significance (Song et al., 2023). Therefore, blob detection performance in our work may only faithfully reflect the conscious level in each monocular pathway, but it is probably not an appropriate index tightly associated with the attentional modulations on monocular responses in early visual areas. Indeed, previous work has argued that attention but not awareness modulates neural activities in V1 during interocular competition (Watanabe et al., 2011), but see (Yuval-Greenberg & Heeger, 2013). We have noticed and discussed the counterintuitive results of blob detection performance in our previous work (Song et al., 2023). Here, with the new counterintuitive finding that inhibition of FEF did not impair the performance of blob detection, we suspect that blob detection performance in the “dichoptic-backward-movie” adaptation paradigm may not be an ideal index that can be used to accurately quantify eye-based attention.

(4) Writing: in general, the manuscript is well written, but clarity should be improved in certain sections.(a) fMRI results: the first sentence is difficult to understand at first read, but it is crucial to understand the results, please reformulate and clarify.

Response: Thanks for the reviewer’s suggestion! In the revised manuscript, we have reformulated this sentence (see page 9 last paragraph or below):

“It was only in the dichoptic condition of experimental runs that participants had to selectively pay more attention to one eye (i.e., eye-based attention). Therefore, we speculate that if certain brain regions exhibit greater activities in the dichoptic condition as compared to the binocular condition in the experimental runs but not in the control runs, the activation of these brain regions could be attributable to eye-based attention.”

(b) Experiment 3: the rationale for experiment one should be straightforward, without a long premise explaining why it would not be necessary.

Response: Thanks for the reviewer’s suggestion! In the revised manuscript, we have streamlined the lengthy premise explaining to make the rationale of Experiment 3 more straightforward (see page 15 last two paragraphs or below):

“The results of Experiment 2 support the notion that eye-based attention was the cause for attention-induced ocular dominance plasticity. However, an alternative account is that the significant two-way interaction between test phase and stimulation site did not stem from any persistent malfunction of FEF in modulating ocular dominance, but rather it was due to some abnormality of binocular rivalry measures in the post-test that occurred after stimulation at the FEF only (and not at the other two brain sites). For instance, stimulation at the FEF might simply reduce the ODI measured in the binocular rivalry post-test.

Therefore, we conducted Experiment 3 to examine how suppression of the three target sites would impact binocular rivalry performance, in case that any unknown confounding factors, which were unrelated to adaptation but related to binocular rivalry measures, contributed to the results.”

(c) Discussion: the language is a bit familiar here and there, a more straightforward style should be preferred (one example: p.19 second paragraph).

Response: Thanks for the reviewer’s suggestion! We have carefully revised the language in the discussion. The discussion following the example paragraph has been largely rewritten.

(5) Minor: the authors might consider using the term "participant" or "observer" instead of "subject" when referring to the volunteers who participated in the study.

Response: Thanks for the reviewer’s suggestion! In the revised manuscript, we have replaced the term “subject” with “participant”.

**Reviewer #3 (Public Review):**
Summary:This study studied the neural mechanisms underlying the shift of ocular dominance induced by "dichoptic-backward-movie" adaptation. The study is self-consistent.Strengths:The experimental design is solid and progressive (relationship among three studies), and all of the raised research questions were well answered.The logic behind the neural mechanisms is solid.The findings regarding the cTMS (especially the position/site can be useful for future medical implications).Weaknesses:Why does the "dichoptic-backward-movie" adaptation matter? This part is severely missing. This kind of adaptation is neither intuitive like the classical (Gbison) visual adaptation, nor practical as adaptation as a research paradigm as well as the fundamental neural mechanism. If this part is not clearly stated and discussed, this study is just self-consistent in terms of its own research question. There are tons of "cool" phenomena in which the neural mechanisms are apparent as "FEF controls vision-attention" but never tested using TMS & fMRI, but we all know that this kind of research is just of incremental implications.

Response: Thanks for the reviewer’s comment! We designed the "dichoptic-backward-movie" adaptation to study the perceptual consequence and mechanisms of sustained attention to a monocular pathway. Since the overall visual input to both eyes during adaptation were identical, any effect (i.e. the change of ocular dominance in our study) after adaptation can be easily ascribed to unbalanced eye-based attention between the two eyes rather than unbalanced input energy across the eyes. In typical short-term monocular deprivation, input signal from one eye is blocked. Accordingly, attention is undoubtedly distributed to the non-deprived eye. The fact that in a short-term monocular deprivation paradigm the deprived eye is also the unattended eye prevents researchers from ascertaining whether unbalanced eye-based attentional allocation contributes to the shift of ocular dominance just like unbalanced visual input across the two eyes. That is why the “dichoptic-backward-movie” adaptation was adopted in the present study. This new paradigm balances the input energy across the eyes but leaves attention unbalanced across the eyes. In the revised manuscript, we have added the description of the “dichoptic-backward-movie” adaptation (see page 3 last paragraph and page 4 first paragraph or below). Hope this complementary information improves the clarity.

“In Song et al. (2023)’s “dichoptic-backward-movie” adaptation paradigm (see Figure 1B), participants are presented with regular movie images in one eye (i.e., attended eye) while the other eye (i.e., unattended eye) received the backward movie images of the same episode. They were also instructed to try their best to follow the logic of the regular movie and ignore the superimposed backward movie. Therefore, the goal-directed eye-based attention was predominantly focused on the attended eye. Song et al. (2023) found that the predominance of the unattended eye in binocular rivalry increased after one hour of adaptation to the “dichoptic-backward-movie”, indicating a shift of perceptual ocular dominance towards the unattended eye. Since the overall energy of visual input from the two eyes was balanced throughout the adaptation period, the change of ocular dominance after adaptation is thought to result from unbalanced eye-based attention rather than unbalanced input energy as in typical short-term monocular deprivation (Bai et al., 2017; Lunghi et al., 2011; Zhou et al., 2014).” In short-term monocular deprivation, input signal from one eye is blocked. Accordingly, attention is biased towards the non-deprived eye. However, it is difficult to tease apart the potential contribution of unbalanced eye-based attention from the consequence of the unbalanced input energy, as the deprived eye is also the unattended eye. Therefore, the advantage of the “dichoptic-backward-movie” adaptation paradigm is to balance the input energy across the eyes but leave attention unbalanced across the eyes.

Our previous work (Song et al., 2023) has shown that eye-based attention plays a role in the formation of ocular dominance shift following adaptation to dichoptic backward movie. However, because the “dichoptic-backward-movie” adaptation paradigm is new, to our knowledge, no literature has ever discovered the brain areas that are responsible for eye-based attention. Our fMRI experiment for the first time resolves this issue, which, we believe, is one of the novelties of the present study. Attention is a pretty general definition of our ability to select limited information for preferential or privileged processing, yet it includes numerous aspects (e.g. spatial attention for spatial locations, feature-based attention for visual features, object-based attention for objects, social attention for social cues, and eye-based attention for monocular pathways etc). Are we 100% sure that the same brain network always underlies every aspect of attention including eye-based attention? No test, no answer. Maybe the answer is Yes, but we are not aware of any evidence for that from literature. It is not unlikely that attention is like an elephant while researchers are like blind people touching the elephant from different angles. Even if all previous researchers have touched the side of the elephant and state that an elephant is no different from a wall, as long as one researcher grabs the elephant’s tail, the “wall” knowledge will be falsified. From this perspective of the essence of science (falsifiable), we have the confidence to say that our fMRI experiment on eye-based attention is novel, because to our knowledge our experiment is the first one to explore the issue. On the basis of the fMRI experiment (otherwise we would have no idea on which precise brain site to apply the cTBS), we could successfully complete the subsequent TMS experiments.

Of course, if the reviewer can kindly point out any previous neuroimaging work we missed that has already disclosed the neural mechanisms underlying human’s eye-based attention, we would truly appreciate the reviewer very much. But even so, we would like to emphasize that the purpose of the current study was actually not to use TMS & fMRI to confirm that “FEF controls visual attention”. As we mentioned in the Abstract and expanded the introduction in the last two paragraphs of Introduction, the goal of the TMS experiments is to examine the causal role of eye-based attention in producing the aftereffect of “dichoptic-backward-movie” adaptation. This research question is also new, thus we do not think the TMS experiments are incremental, either. Our findings provided direct causal evidence for the effect of FEF on modulating ocular dominance through eye-based attention. Please see the last two sentences in the first paragraph on page 20 in the revised manuscript or below,

“Interestingly, in our Experiment 2 this aftereffect was significantly attenuated after we temporarily inhibited the cortical function of FEF via cTBS. This finding indicates the crucial role of FEF in the formation of attention-induced ocular dominance shift.”

as well as the last sentence of the Abstract,

“…and in this network, FEF plays a crucial causal role in generating the attention-induced ocular dominance shift.”

**Recommendations for the authors:**

**Reviewer #1 (Recommendations For The Authors):**
(1) The hemispheric asymmetry in the eye-based attention-related cortex should be further examined and discussed. For example, IPS in both hemispheres was identified in the fMRI experiment. It is not clear why only the right IPS was stimulated in the TMS experiment.

Response: Thanks for the comment. We have elucidated the reasons for the experimental design with hemispheric asymmetry in FEF and IPS. Please see our response to the Weakness #1 raised by Reviewer #1 in the Public Review section.

(2) It is known that the frontoparietal cortex plays a role in the contralateral shift of attentional allocation. Meanwhile, the latest stage of ocular-specific representation is V1. The authors should discuss how the eye-related function can be achieved in FEF.

Response: Thanks for the comment. we have discussed how FEF regulates attention-induced ocular dominance shift (see page 21 second paragraph to page 23 first paragraph in the revised manuscript, and our response to the Weakness #2 raised by Reviewer #1 in the PublicReview section).

(3) To further validate the role of FEF in eye-related attention shifts, the authors may consider using the traditional monocular deprivation paradigm with fMRI and TMS. It would be valuable to compare the neural mechanisms related to the classical monocular deprivation paradigm with the current findings.

Response: Thanks for the reviewer’s suggestion! That is indeed an interesting research topic that we are currently exploring. The current study investigated the attention-induced ocular dominance shift with the “dichoptic-backward-movie-adaptation” paradigm. This paradigm is substantially different from traditional short-term monocular deprivation. In our Neuroscience Bulletin paper (Song et al. 2023), we discuss the reason as follows.

“An alternative account of our results is the homeostatic plasticity mechanism. The function of this mechanism is to stabilize neuronal activity and prevent the neuronal system from becoming hyperactive or hypoactive. For this goal, the mechanism moves the neuronal system back toward its baseline after a perturbation [51, 52]. In our case, the aftereffect can be explained such that the visual system boosts the signals from the unattended eye to maintain the balance of the network’s excitability. However, this account cannot easily explain why the change of neural ocular dominance led by prolonged eye-based attention was observed here using the binocular rivalry testing stimuli, but absent in the previous research using the binocularly fused stimuli [11]. In contrast, a recent SSVEP study also using the binocularly fused stimuli has successfully revealed a shift of neural ocular dominance after two hours of monocular deprivation [31], which is in line with the homeostatic plasticity account. Therefore, the mechanisms underlying the “dichoptic-backward-movie” adaptation and monocular deprivation are probably not fully overlapped with each other; and the binocular rivalry mechanism described in the ocular-opponency-neuron model seems to be more preferable than the homeostatic plasticity mechanism in accounting for the present findings.”

Therefore, before asking whether FEF plays a role in the attention-induced ocular dominance shift in a traditional monocular deprivation paradigm, one should probably first examine whether attention also plays a role in traditional monocular deprivation, and whether the ocular-opponency-neuron adaptation account can also be used to explain the traditional monocular deprivation effect. Our newly accepted paper “Negligible contribution of adaptation of ocular opponency neurons to the effect of short-term monocular deprivation” (https://www.frontiersin.org/articles/10.3389/fpsyg.2023.1282113/full) gives a generally negative answer to the second question. And as to the first question, we have one manuscript under review and another ongoing study. In other words, to get a satisfactory answer to this particular comment of this reviewer, we need to first obtain clear answers to the two above questions. We think this is far beyond the scope of one single manuscript.

(4) The authors only presented regular movies to the dominant eye to maximize the ocular dominance shift. This critical information of design should be clarified, not only in the method section.

Response: Thanks for the reviewer’s suggestion! In the Results section of Experiment 2, we have added a description of this critical information of design (see page 11 last paragraph to page 12 first paragraph or below):

“Then, participants adapted to the “dichoptic-backward-movie” in which regular movie images were presented to the dominant eye to maximize the effect of eye dominance shift (Song et al., 2023). Meanwhile they were asked to detect some infrequent blob targets presented on the movie images in one eye at the same time.”

(5) The frame rate of the movie is 30 fps, which is much lower than a typical 60 fps visual presentation, does this have an effect on the adaptation outcome?

Response: To our best of knowledge, there is no evidence that the frame rate of the movie influences the aftereffect of attention-induced ocular dominance shift. In our previous research, the frame rate of the movie during adaptation was 25 fps, which still produced a stable adaptation aftereffect (Song et al., 2023). And the frame rate of the movie was 30 fps in our monocular deprivation work (Lyu et al., 2020), which showed a similar monocular deprivation effect we previously observed in an altered reality study (Bai et al., 2017). The frame rate of the altered-reality video in Bai et al.’s (2017) work was 60 fps. All these clues suggest that the frame rate does not have an effect on the adaptation outcome.

(6) Figure 5: The ODSE derived from ODI in Experiment 3 should also be illustrated, for a better comparison with results from Experiment 2.

Response: Thanks for the reviewer’s suggestion! In the revised manuscript, we have added the results of ODSE in Experiment 3 to Figure 5 (see page 15 or below):

**Author response image 1. sa4fig1:** The results of (**A**) the ocular dominance index (ODI), (**B**) the ocular dominance shift effects (ODSE) in Experiment 2, (**C**) the ODI and (**D**) the ODSE in Experiment 3. The bars show the grand average data for each condition. The individual data are plotted with gray lines or dots. The dashed gray line represents the absolute balance point for the two eyes (ODI = 0.5). Error bars indicate standard errors of means. * p < .05; ** p < .01; n.s. p > .05.

(7) Spelling issues: "i.e." → "i.e.,"

Response: Thanks for the reviewer’s suggestion! In the revised manuscript, we have changed “i.e.” to “i.e.,”.

**Reviewer #2 (Recommendations For The Authors):**
Linked to weakness 3: Ideally, a control experiment with cTBS and dichoptic stimulation without sound but with the blob discrimination task should be performed to be able to make important claims about the neural mechanisms involved in eye-based attention.

Response: Thanks for the comment. We have performed a new experiment as the reviewer suggested. Please see our response to the Weakness #3 raised by Reviewer #2 in the Public Review section.

**Reviewer #3 (Recommendations For The Authors):**
(1) The neural mechanisms are so apparent. We all know the FEF\IPS\SC matter in vision and attention and gaze. This is not groundbreaking.

Response: As we addressed in our response to Reviewer #3’s public comment, the current study aimed at investigating the causal mechanism for eye-based attentional modulation of ocular dominance plasticity rather than simply the role of FEF\IPS\SC in visual attention. Moreover, eye-based attention is a less investigated aspect of visual attention. The neural mechanism underlying eye-based attention is still largely unknown, and seeking the brain areas for controlling eye-based attention is the necessary preparation work for applying the cTBS. We have responded in detail to Reviewer #3’s public comment why we think both the fMRI and TMS experiments are novel to the field, which we will not reiterate it here to avoid redundancy.

(2) Why does the "dichoptic-backward-movie" adaptation matter? Is playing a backward movie to one eye realistic? Does that follow the efficient coding? Is that a mere consequence of information theory?

Response: Thanks for the comments. We have added the description of the “dichoptic-backward-movie” adaptation paradigm in the revised manuscript (see page 3 last paragraph and page 4 first paragraph or our response to this reviewer’s Public comment).

Is it realistic to play backward movie to one eye? We feel this question is somehow ambiguous to us. If the reviewer means the technical operability for such stimulus presentation, we can assure it since we have used this paradigm in both the current and previously published studies. To be more specific, we made the video stimuli in advance. The left half of the video was the regular movie and the right half was the backward version of the same movie (or vice versa). When viewing such video stimuli through stereoscopes, participants could only see the left half of the video with the left eye and the right half of the video with the right eye. In other words, the regular movie and backward movie were viewed dichoptically. Alternatively, if the reviewer means that such dichoptic presentation rarely happens in real world thus not realistic, we agree with the reviewer on one hand. On the other hand, we have explained on page 3 last paragraph and page 4 first paragraph why it is a particular useful paradigm for the main purpose of the present study. Let us make a similar example. The phenomenon of binocular rivalry rarely happens in everyday life. So people may say binocular rivalry is not realistic. However, our visual system does have the ability to deal with such conflicting visual inputs across the eyes, even binocular rivalry is unrealistic! Sometimes it is fun to investigate those seemingly unrealistic functions of our brains since those may also reveal the mystery of our neural system. As we know, despite binocular rivalry is uncommon in daily life, it is frequently used to investigate awareness. And in our work, we use binocular rivalry to measure perceptual ocular dominance.

Finally, the reviewer queried about if the "dichoptic-backward-movie" adaptation paradigm follow efficient coding and information theory. The information theory and efficient coding assume that messages with low expectedness or of rare occurrence would attract more attention and induce larger neural responses than those with high expectedness. In the "dichoptic-backward-movie" adaptation paradigm, the backward movie should be less expected since the actions of the characters in the backward movie appeared illogical. Thus, according to the information theory and efficient coding, it would be expected that more attention was paid to the backward movie and thus the backward movie might dominate the awareness for a longer period during adaptation (Zhang et al., 2012). However, we instructed participants to follow the regular movie during adaptation. The results of blob detection task also showed a better task performance when the targets appeared in the eye presented with the regular movie, which contradicted with the prediction of the information theory and efficient coding. Thus, it seems not very likely that the "dichoptic-backward-movie" adaptation followed efficient coding and information theory.

References

Bai, J., Dong, X., He, S., & Bao, M. (2017). Monocular deprivation of Fourier phase information boosts the deprived eye’s dominance during interocular competition but not interocular phase combination. Neuroscience, 352, 122-130. https://doi.org/10.1016/j.neuroscience.2017.03.053

Benjamini, Y., & Hochberg, Y. (1995). Controlling the false discovery rate: a practical and powerful approach to multiple testing. Journal of the Royal statistical society: series B (Methodological), 57(1), 289-300. https://doi.org/10.1111/j.2517-6161.1995.tb02031.x

Choe, E., & Kim, M.-S. (2022). Eye-specific attentional bias driven by selection history. Psychonomic Bulletin & Review, 29(6), 2155-2166. https://doi.org/10.3758/s13423-022-02121-0

Corbetta, M., & Shulman, G. L. (2002). Control of goal-directed and stimulus-driven attention in the brain. Nature reviews neuroscience, 3(3), 201-215. https://doi.org/10.1038/nrn755

Dong, X., Gao, Y., Lv, L., & Bao, M. (2016). Habituation of visual adaptation. Sci Rep, 6, 19152. https://doi.org/10.1038/srep19152

Duecker, F., Formisano, E., & Sack, A. T. (2013). Hemispheric differences in the voluntary control of spatial attention: direct evidence for a right-hemispheric dominance within frontal cortex. Journal of Cognitive Neuroscience, 25(8), 1332-1342. https://doi.org/10.1162/jocn_a_00402

Esterman, M., Liu, G., Okabe, H., Reagan, A., Thai, M., & DeGutis, J. (2015). Frontal eye field involvement in sustaining visual attention: evidence from transcranial magnetic stimulation. Neuroimage, 111, 542-548. https://doi.org/10.1016/j.neuroimage.2015.01.044

Gallotto, S., Schuhmann, T., Duecker, F., Middag-van Spanje, M., de Graaf, T. A., & Sack, A. T. (2022). Concurrent frontal and parietal network TMS for modulating attention. iScience, 25(3), 103962. https://doi.org/10.1016/j.isci.2022.103962

Lega, C., Ferrante, O., Marini, F., Santandrea, E., Cattaneo, L., & Chelazzi, L. (2019). Probing the neural mechanisms for distractor filtering and their history-contingent modulation by means of TMS. Journal of Neuroscience, 39(38), 7591-7603. https://doi.org/10.1523/JNEUROSCI.2740-18.2019

Lunghi, C., Burr, D. C., & Morrone, C. (2011). Brief periods of monocular deprivation disrupt ocular balance in human adult visual cortex. Curr Biol, 21(14), R538-539. https://doi.org/10.1016/j.cub.2011.06.004

Lyu, L., He, S., Jiang, Y., Engel, S. A., & Bao, M. (2020). Natural-scene-based Steady-state Visual Evoked Potentials Reveal Effects of Short-term Monocular Deprivation. Neuroscience, 435, 10-21. https://doi.org/10.1016/j.neuroscience.2020.03.039

Mayrhofer, H. C., Duecker, F., van de Ven, V., Jacobs, H. I., & Sack, A. T. (2019). Hemifield-specific correlations between cue-related blood oxygen level dependent activity in bilateral nodes of the dorsal attention network and attentional benefits in a spatial orienting paradigm. Journal of Cognitive Neuroscience, 31(5), 625-638. https://doi.org/10.1162/jocn_a_01338

Rezec, A., Krekelberg, B., & Dobkins, K. R. (2004). Attention enhances adaptability: evidence from motion adaptation experiments. Vision Res, 44(26), 3035-3044. https://doi.org/10.1016/j.visres.2004.07.020

Sack, A. T. (2010). Using non-invasive brain interference as a tool for mimicking spatial neglect in healthy volunteers. Restorative neurology and neuroscience, 28(4), 485-497. https://doi.org/10.3233/RNN-2010-0568

Said, C. P., & Heeger, D. J. (2013). A model of binocular rivalry and cross-orientation suppression. PLoS computational biology, 9(3), e1002991. https://doi.org/10.1371/journal.pcbi.1002991

Song, F., Lyu, L., Zhao, J., & Bao, M. (2023). The role of eye-specific attention in ocular dominance plasticity. Cerebral Cortex, 33(4), 983-996. https://doi.org/10.1093/cercor/bhac116

van den Bergh, D., Wagenmakers, E.-J., & Aust, F. (2023). Bayesian Repeated-Measures Analysis of Variance: An Updated Methodology Implemented in JASP. Advances in Methods and Practices in Psychological Science, 6(2), 25152459231168024. https://doi.org/10.1177/25152459231168024

van Doorn, J., van den Bergh, D., Böhm, U., Dablander, F., Derks, K., Draws, T., Etz, A., Evans, N. J., Gronau, Q. F., Haaf, J. M., Hinne, M., Kucharský, Š., Ly, A., Marsman, M., Matzke, D., Gupta, A., Sarafoglou, A., Stefan, A., Voelkel, J. G., & Wagenmakers, E. J. (2021). The JASP guidelines for conducting and reporting a Bayesian analysis. Psychonomic Bulletin & Review, 28(3), 813–826. https://doi.org/10.3758/s13423-020-01798-5

Wagenmakers, E. J., Love, J., Marsman, M., Jamil, T., Ly, A., Verhagen, J., Selker, R., Gronau, Q. F., Dropmann, D., Boutin, B., Meerhoff, F., Knight, P., Raj, A., van Kesteren, E. J., van Doorn, J., Šmíra, M., Epskamp, S., Etz, A., Matzke, D., de Jong, T., van den Bergh, D., Sarafoglou, A., Steingroever, H., Derks, K., Rouder, J. N., & Morey, R. D. (2018). Bayesian inference for psychology. Part II: Example applications with JASP. Psychonomic Bulletin & Review, 25(1), 58–76. https://doi.org/10.3758/s13423-017-1323-7

Watanabe, M., Cheng, K., Murayama, Y., Ueno, K., Asamizuya, T., Tanaka, K., & Logothetis, N. (2011). Attention but not awareness modulates the BOLD signal in the human V1 during binocular suppression. Science, 334(6057), 829-831. https://doi.org/10.1126/science.1203161

Wong, S. P., Baldwin, A. S., Hess, R. F., & Mullen, K. T. (2021). Shifting eye balance using monocularly directed attention in normal vision. J Vis, 21(5), 4. https://doi.org/10.1167/jov.21.5.4

Yuval-Greenberg, S., & Heeger, D. J. (2013). Continuous flash suppression modulates cortical activity in early visual cortex. J Neurosci, 33(23), 9635-9643. https://doi.org/10.1523/jneurosci.4612-12.2013

Zhang, P., Jiang, Y., & He, S. (2012). Voluntary attention modulates processing of eye-specific visual information. Psychol Sci, 23(3), 254-260. https://doi.org/10.1177/0956797611424289

Zhou, J., Reynaud, A., & Hess, R. F. (2014). Real-time modulation of perceptual eye dominance in humans. Proc Biol Sci, 281(1795). https://doi.org/10.1098/rspb.2014.1717